# Extreme Memorization via Scale of Initialization

**Harsh Mehta**
Google Research
harshm@google.com

**Ashok Cutkosky**
Boston University
ashok@cutkosky.com

**Behnam Neyshabur**
Blueshift, Alphabet
neyshabur@google.com

## Abstract

We construct an experimental setup in which changing the scale of initialization strongly impacts the implicit regularization induced by SGD, interpolating from good generalization performance to completely memorizing the training set while making little progress on the test set. Moreover, we find that the extent and manner in which generalization ability is affected depends on the activation and loss function used, with $\sin$ activation demonstrating extreme memorization. In the case of the homogeneous ReLU activation, we show that this behavior can be attributed to the loss function. Our empirical investigation reveals that increasing the scale of initialization correlates with misalignment of representations and gradients across examples in the same class. This insight allows us to devise an alignment measure over gradients and representations which can capture this phenomenon. We demonstrate that our alignment measure correlates with generalization of deep models trained on image classification tasks.

## 1 Introduction

Training highly overparametrized deep neural nets on large datasets has been a very successful modern recipe for building machine learning systems. As a result, there has been a significant interest in explaining some of the counter-intuitive behaviors seen in practice, with the end-goal of further empirical success.

One such counter-intuitive trend is that the number of parameters in models being trained have increased considerably over time, and yet these models continue to increase in accuracy without loss of generalization performance. In practice, improvements can be observed even after the point where the number of parameters far exceed the number of examples in the dataset, i.e., when the network is overparametrized (Zhang et al., 2016; Arpit et al., 2017) . These wildly over-parameterized networks avoid overfitting even without explicit regularization techniques such as weight decay or dropout, suggesting that the training procedure (usually SGD) has an implicit bias which encourages the net to generalize (Caruana et al., 2000; Neyshabur et al., 2014; 2019; Belkin et al., 2018a; Soudry et al., 2018).

**Contributions** In order to understand the interplay between training and generalization, we investigate situations in which the network can be made to induce a scenario in which the accuracy on the test set drops to random chance while maintaining perfect accuracy on the training set. We refer to this behavior as *extreme memorization*, distinguished from the more general category of memorization where either test set performance is higher than random chance or the net fails to attain perfect training set accuracy. In this paper, we examine the effect of *scale of initialization* on the generalization performance of SGD. We found that it is possible to construct an experimental setup in which simply changing the scale of the initial weights allows for a continuum of generalization ability, from very little overfitting to perfectly memorizing the training set. It is our hope that these observations provide fodder for further advancements in both theoretical and empirical understanding of generalization. [1]

---

[1]The code used for experiments is open-sourced at https://github.com/google-research/google-research/tree/master/extreme_memorization

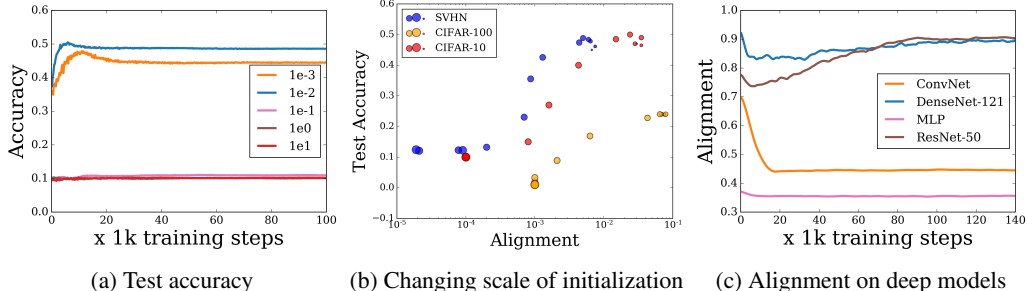

(a) Test accuracy   (b) Changing scale of initialization   (c) Alignment on deep models

Figure 1: (a) Results when using sin activation function in a 2-layer MLP. We initialize the first layer using random normal distribution with mean zero and vary the standard deviation $\sigma$ as shown in the plots. Initialization scheme for the top layer is kept unchanged and uses a glorot uniform initializer (Glorot & Bengio, 2010). The plot shows the drastic changes in generalization ability solely due the changes in scaling on CIFAR-10 dataset. Plot (b) shows the correlation between best test accuracy and gradient alignment values across 3 different datasets, CIFAR-10 (Krizhevsky, 2009) CIFAR-100 and SVHN as we change scale of initialization. Finally, plot (c) illustrates that the alignment measure can also capture differences in generalization across model architectures. Note that, in order to do a fair comparison, all hyperparameters (e.g. learning rate, optimizer) are kept constant.

- We construct a two-layer feed forward network using sin activation and observe that increasing the scale of initialization of the first layer has a strong effect on the implicit regularization induced by SGD, approaching extreme memorization of the training set as the scale is increased. We observe this phenomenon on 3 different image classification datasets: CIFAR-10, CIFAR-100 and SVHN.
- For the popular ReLU activation, one might expect that changing the scale should not affect the predictions of network, due to its homogeneity property. Nevertheless, even with ReLU activation we see a similar drop in generalization performance. We demonstrate that generalization behavior can be attributed further up in the network to a variety of common loss functions (softmax cross-entropy, hinge and squared loss).
- Gaining insight from these phenomena, we devise an empirical "gradient alignment" measure which quantifies the agreement between gradients for examples corresponding to a class. We observe that this measure correlates well with the generalization performance as the scale of initialization is increased. Moreover, we formulate a similar notion for representations as well.
- Finally, we provide evidence that our alignment measure is able to capture generalization performance across architectural differences of deep models on image classification tasks.

## 2 RELATED WORK

Understanding the generalization performance of neural networks is a topic of widespread interest. While overparametrized nets generalize well when trained via SGD on real datasets, they can just as easily fit the training data when the labels are completely shuffled (Zhang et al., 2016). In fact, Belkin et al. (2018b) show that the perfect overfitting phenomenon seen in deep nets can also be observed in kernel methods. Further studies like Neyshabur et al. (2017); Arpit et al. (2017) expose the qualitative differences between nets trained with real vs random data. Generalization performance has been shown to depend on many factors including model family, number of parameters, learning rate schedule, explicit regularization techniques, batch size, etc (Keskar et al., 2016; Wilson et al., 2017). Xiao et al. (2019) further characterize regions of hyperparameter spaces where the net memorizes the training set but fails to generalize completely.

Interestingly, there has been recent work showing that over-parametrization aids not just with generalization but optimization too (Du et al., 2019; 2018; Allen-Zhu et al., 2018; Zou et al., 2019). Du et al. (2018) show that for sufficiently over-parameterized nets, the gram matrix of the gradients induced by ReLU activation remains positive definite throughout training due to parameters staying close to initialization. Moreover, in the infinite width limit the network behaves like its linearized version of the same net around initialization (Lee et al., 2019). Jacot et al. (2018) explicitly characterize the solution obtained by SGD in terms of Neural Tangent Kernel which, in the infinite

width limit, stays fixed through the training iterations and deterministic at initialization. Finally, Frankle & Carbin (2018); Frankle et al. (2019) hypothesize that overparametrized neural nets contain much smaller sub-networks, called "lottery tickets", which when trained in isolation can match the performance of the original net.

From an optimization standpoint, several initialization schemes have been proposed in order to facilitate neural network training (Glorot & Bengio, 2010; He et al., 2015a). Balduzzi et al. (2017) identify the *shattered gradient problem* where the correlation between gradients w.r.t to the input in feedforward networks decays exponentially with depth. They further introduce *looks linear* initialization in order to prevent shattering. Recent work explores some intriguing behavior induced by changing just the scaling of the net at initialization. Building on observation made by others (Li & Liang, 2018; Du et al., 2019; 2018; Zou et al., 2019; Allen-Zhu et al., 2018), Chizat & Bach (2018) formally introduce the notion of *lazy training*, a phenomenon in which an over-parametrized net can converge to zero training loss even as parameters barely change. Chizat & Bach (2018) further observe that any model can be pushed to this regime by scaling the initialization by a certain factor, assuming the output is close to zero at initialization. Moreover, Woodworth et al. (2020) expand on how scale of initialization acts as a controlling quantity for transitioning between two very different regimes, called the kernel and rich regimes. In the kernel regime, the behavior of the net is equivalent to learning using kernel methods, while in the rich regime, gradient descent shows richer inductive biases which are not captured by RKHS norms. In practice, the transition from rich regime to kernel regime also comes with a drop in generalization performance. Geiger et al. (2019) further explore the interplay between hidden layer size and scale of initialization in disentangling both regimes, specifically that the scale of initialization, which separates kernel and rich regime, is a function of the hidden size.

On a somewhat orthogonal direction, from a theoretical perspective, several studies attempt to bound the generalization error of the network based on VC-dimension (Vapnik, 1971), sharpness based measures such as PAC-Bayes bounds (McAllester, 1999; Dziugaite & Roy, 2017; Neyshabur et al., 2017), or norms of the weights (Bartlett, 1998; Neyshabur et al., 2015b; Bartlett et al., 2017; Neyshabur et al., 2019; Golowich et al., 2019). Further works explore generalization from an empirical standpoint such as sharpness based measures (Keskar et al., 2016), path norm (Neyshabur et al., 2015a) and Fisher-Rao metric (Liang et al., 2017). A few have also emphasized the role of distance from initialization in capturing generalization behavior (Dziugaite & Roy, 2017; Nagarajan & Kolter, 2019; Neyshabur et al., 2019; Long & Sedghi, 2019).

Li & Liang (2018) study 2-layer ReLU net and points out that final learned weights are accumulated gradients added to the random initialization and these accumulated gradients have low rank when trained on structured datasets. Wei & Ma (2019) obtain tighter bounds by considering data-dependent properties of the network such as norm of the Jacobians of each layer with respect to the previous layers. More recently, Chatterjee (2020) hypothesize that similar examples lead to similar gradients, reinforcing each other in making the the overall gradient stronger in these directions and biasing the net to make changes in parameters which benefit multiple examples.

## 3 EXTREME MEMORIZATION

In this section, we discuss the experimental setup which leads to extreme memorization due to increase in scale of initialization. To reiterate, we refer to the scenario where the net obtains perfect training accuracy but random chance performance on the test set as **extreme memorization**.

In order to investigate this in the simplest setup possible, we consider a 2-layer feed-forward network trained using stochastic gradient descent (SGD):

$$z(x) = W_2\phi(W_1 x)$$

where $\phi$ is the chosen activation function, $\mathbf{x} \in \mathbb{R}^p$, $\mathbf{W}_1 \in \mathbb{R}^{h \times p}$, $\mathbf{W}_2 \in \mathbb{R}^{k \times h}$ and $\mathbf{z} \in \mathbb{R}^k$ is the output of the net. The aim is to find parameters $[\mathbf{W}_1^*, \mathbf{W}_2^*]$ which minimizes the empirical loss $\mathcal{L} = \frac{1}{n}\sum_{i=1}^{n} \ell(z(\mathbf{x}_i), \mathbf{y}_i)$ given i.i.d draws of $n$ data points $\{(\mathbf{x}_i, \mathbf{y}_i)\}$ from some unknown joint distribution over $\mathbf{x} \in \mathbb{R}^p$ and $\mathbf{y} \in \mathbb{R}^k$. We focus on multi-class classification problems, in which each $\mathbf{y}$ is restricted to be one of the standard basis vectors in $\mathbb{R}^k$. We use the notations $\ell_i = \ell(\mathbf{z}(\mathbf{x}_i), \mathbf{y}_i)$ and $\mathbf{r}_i = \phi(\mathbf{W}_1 \mathbf{x}_i)$ is a shorthand for the hidden layer representation for input $\mathbf{x}_i$. Also, for any $c \in \{1, \ldots, k\}$, we use the shorthand $y = c$ to say that $\mathbf{y}$ is the $c$th standard basis vector.

In our experiments, we choose a large hidden size so that the net is very over-parameterized and always gets perfect accuracy on the training set. Also, since we are only interested in studying the implicit regularization induced by SGD, we do not use explicit regularizers like weight decay, dropout, etc. More details on the exact setup, datasets used and hyper-parameters are in the appendix.

## 3.1 SIN ACTIVATION

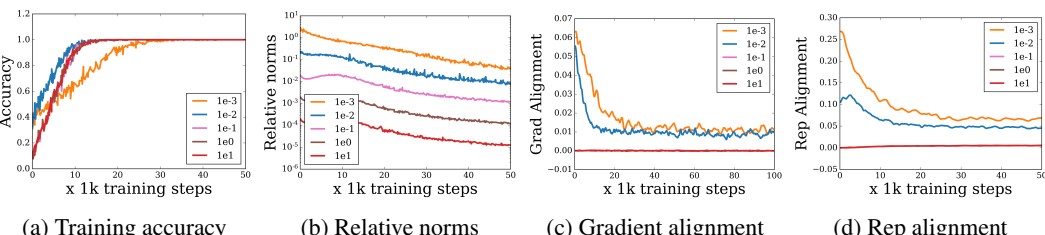

| (a) Training accuracy | (b) Relative norms | (c) Gradient alignment | (d) Rep alignment |

Figure 2: Results when using sin activation function in a 2-layer MLP applied on CIFAR-10 dataset (Krizhevsky, 2009). We initialize $W_1$ using random normal distribution with mean zero and vary the standard deviation $\sigma$ as shown in the plots. The initialization scheme for $W_2$ is kept unchanged, using a Glorot uniform initializer (Glorot & Bengio, 2010). (a) shows the the evolution and rate of attaining perfect training accuracy. (b) plots the norm of the gradients of $W_1$ over norm of $W_1$. As elucidated in (Chizat & Bach, 2018), increasing the scale initialization leads to gradients being increasingly smaller than the weights and thus weights not being able to move very far from initialization. (c) shows how example gradient alignment can capture differences in generalization ability in case of sin activation as the scale of initialization is increased. Plot (d) shows that representation alignment is also able to discriminate generalization ability induced at high scale of initialization. We obtain similar results on CIFAR-100 and SVHN datasets as well, which are included in the appendix.

As shown in Figure 1, setting $\phi$ to sin function results in a degradation of generalization performance to the point of extreme memorization just by increasing the scale of initialization of the hidden layer $W_1$. Intuitively, when using sin activations, if $W_1$ remains close to its initial value, then a single hidden layer can be approximated by a kernel machine with a specific shift-invariant kernel $K$, where $K$ is determined by the initializing distribution (Rahimi & Recht, 2008). For example, when the initializing distribution is a Gaussian with standard deviation $\sigma$, $K$ is a Gaussian kernel with width $1/\sigma$. Formally, consider a network architecture of $z(x) = W_2\phi(W_1 x + b)$, where $W_1$ is a matrix whose entries are initialized via a Gaussian distribution with variance $\sigma^2$ and $b \in \mathbb{R}^h$ is a bias vector whose coordinates are initialized uniformly from $[0, 2\pi]$. Then (Rahimi & Recht, 2008) showed

$$\mathop{\mathbb{E}}_{W_1, b}[\langle \phi(W_1 x + b), \phi(W_1 x' + b) \rangle] \propto \exp\left(-\frac{\sigma^2 \|x - x'\|^2}{2}\right) \tag{1}$$

Thus, when holding $W_1$ and $b$ fixed, the network approximates a kernel machine with a Gaussian kernel whose width decreases as $W_1$ is scaled up (which corresponds to increasing the variance parameter in its initialization). In this scenario, one expects that the classifier will obtain near-perfect accuracy on the train data, but have no signal elsewhere because all points are nearly orthogonal in the kernel space. We did not specify a bias vector in our architecture, but intuitively one expects similar behavior. In fact, we have the following analogous observation (proved in Appendix B):

**Theorem 1.** *Suppose each entry of $W_1$ is initialized via a Gaussian with mean 0 and variance $\sigma^2$. Then for any $x$ and $x'$, we have*

$$\left| \mathop{\mathbb{E}}_{W_1}[\langle \phi(W_1 x), \phi(W_1 x') \rangle] \right| \leq h \exp\left(-\frac{\sigma^2 \|x - x'\|^2}{2}\right)$$

This suggests that for large enough $\sigma$, the vectors $\phi(W_1 x)$ will be nearly uncorrelated in expectation at initialization. Further, for any loss function $\ell$ and label $y$, we have that the columns of $\nabla_{W_2} \ell(z(x), y)$ are proportional to $\phi(W_1 x)$, and so these gradients should also display a lack of correlation as $\sigma$ increases. We argue that this lack of correlation leads to memorization behavior. By *memorization*, we mean that our trained model will have near-perfect accuracy on the training set, while having very low or even near-random performance on the testing set, indicating that the model has "memorized" the training set without learning anything about the testing set.

To gain some intuition for why we might expect poor correlation among features or gradients to produce memorization, let us take a look at an extreme case where the gradients for all the examples

are orthogonal to each other. More concretely, suppose the true data distribution is such that for all independent samples $(x_1, y_1), (x_2, y_2)$ with $(x_1, y_1) \neq (x_2, y_2)$, we have $\langle \nabla \ell_1, \nabla \ell_2 \rangle < \epsilon$ for all $W_1, W_2$ for some small $\epsilon$. Then we should expect that taking a gradient step along any given example gradient should have a negligible $O(\epsilon)$ effect on the loss for any other example. As a result, the final trained model may achieve very small loss on the training set, but should learn essentially nothing about the test set - it will be a perfectly memorizing model.

## 3.2 MEASURING ALIGNMENT

Motivated by this orthogonality intuition, we wish to develop an empirical metric that can measure the degree to which training points are well-aligned with each other. We begin with a review of related metrics in the existing literature and suggest improvements in order to better capture generalization.

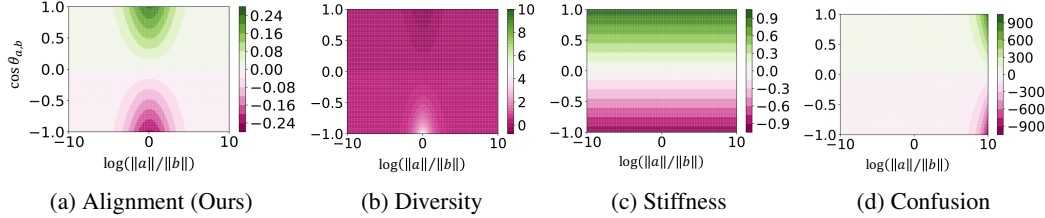

| (a) Alignment (Ours) | (b) Diversity | (c) Stiffness | (d) Confusion |

Figure 3: Comparing different gradient-based measures for the simple case of having two samples from the same class where $a = \nabla \ell_1$ and $b = \nabla \ell_2$.

**Related statistics** Other relevant gradient-based measures have been suggested for understanding optimization or generalization. One such measure is Gradient diversity (Yin et al., 2018), that quantifies the extent to which individual gradients are orthogonal to each other and is defined as $\sum_{i=1}^{n} \|\nabla \ell_i\|_2^2 / \|\sum_{i=1}^{n} \nabla \ell_i\|_2^2$. Unfortunately, Gradient Diversity did not correlate with generalization in our experiments in Section 3. Moreover, as shown in Figure 3, Gradient Diversity is most sensitive when the cosine of the angle between two gradient is highly negative, a scenario which is rare in high dimensional spaces. Furthermore, this notation does not take the class information into account and treats all pairs of samples equally. Cosine Gradient Stiffness (Fort et al., 2019) is another measure to capture the similarity of gradients and can be calculated as $\mathbb{E}_{i \neq j}[\cos(\nabla \ell_i, \nabla \ell_j)]$. Fort et al. (2019) also define a modified version of Cosine Gradient Stiffness that allows this calculation within classes. Although it measures a quantity which is close to what we want, as shown in Figure 3, this measure is invariant to the scale of the gradient. That means that samples with very small gradients would be weighted as much as samples with large gradients, thus discarding valuable information. Finally, we also consider Gradient Confusion (Sankararaman et al., 2019), which can be calculated as $\min_{i \neq j} \langle \nabla \ell_i, \nabla \ell_j \rangle$. We note that, as shown in Figure 3, gradient confusion is sensitive to the norm of gradients and is most affected by the ratio of the norms. Also, similar to Gradient Diversity, this measure does not take the class information into account.

With these observations in mind, we formulate our measure of alignment $\Omega$ between gradient vectors and compare it with other measures in Figure 3. Note that we normalize our alignment measure by the **mean** gradient norm in order to avoid discarding magnitude information from individual gradients:

$$\Omega := \frac{\mathbb{E}_{i \neq j}[\langle \nabla \ell_i, \nabla \ell_j \rangle]}{\mathbb{E}[\|\nabla \ell\|]^2} \tag{2}$$

Assuming $n$ vectors, we can further expand since $\mathbb{E}_{i \neq j}[\langle \nabla \ell_i, \nabla \ell_j \rangle] = \frac{\sum_{i \neq j} \langle \nabla \ell_i, \nabla \ell_j \rangle}{n(n-1)}$ and $\mathbb{E}[\|\nabla \ell\|] = \frac{\sum_{i=1}^{n} \|\nabla \ell_i\|}{n}$

$$\Omega = \frac{n \sum_{i \neq j} \langle \nabla \ell_i, \nabla \ell_j \rangle}{(n-1)(\sum_{i=1}^{n} \|\nabla \ell_i\|)^2} \tag{3}$$

**Efficient computation of alignment** Note that $\sum_{i \neq j} \langle \nabla \ell_i, \nabla \ell_j \rangle$ may appear to require $O(n^2)$ time to compute, but in fact it can be computed in $O(n)$ time by reformulating the expression as:

$$\sum_{i \neq j} \langle \nabla \ell_i, \nabla \ell_j \rangle = \left\| \sum_{i=1}^{n} \nabla \ell_i \right\|^2 - \sum_{i=1}^{n} \|\nabla \ell_i\|^2 \tag{4}$$

For comparison, Gradient Diversity also can be computed in $O(n)$ time but Cosine Gradient Stiffness and Gradient Confusion appears to require $O(n^2)$ time.

**Alignment within a class** In order to compute alignment for a specific class, we propose selecting the examples of a class and compute the alignment for that subset. More specifically, we formulate specific alignment for each class $c = 1, \ldots, k$, as follows:

$$\Omega_c := \frac{n_c \sum_{i \neq j} \langle \nabla \ell_i, \nabla \ell_j \rangle \mathbb{1}[y_i = y_j = c]}{(n_c - 1)(\sum_i^n \|\nabla \ell_i\| \mathbb{1}[y_i = c])^2} \tag{5}$$

where $n_c$ is the number of training examples with label $y = c$ and $\mathbb{1}[p]$ is the indicator of the proposition $p$ - it is one if $p$ is true and zero otherwise. We further take the mean of $\Omega_c$ over all classes for an overall view of how in-class alignment behaves.

$$\Omega_{in-class} := \frac{1}{k} \sum_{c=1}^{k} \Omega_c \tag{6}$$

As shown in Figure 1, $\Omega_{in-class}$ correlates well with generalization ability of the net when scale of initialization is increased. **All of our gradient alignment plots report the average in-class alignment $\Omega_{in-class}$.**

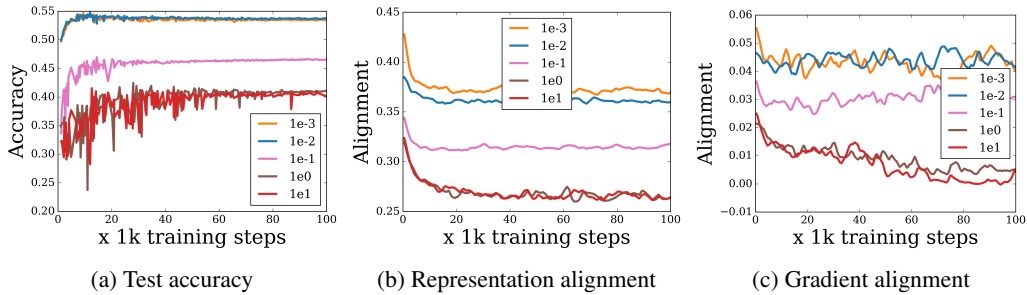

(a) Test accuracy        (b) Representation alignment        (c) Gradient alignment

Figure 4: Results when using ReLU activation in a 2-layer MLP with Softmax cross-entropy loss function when trained on CIFAR-10 dataset. Similar to Figure 2, $W_1$ is initialized with random normal distribution with mean zero and varying standard deviation scale $\sigma$ as shown in the plots. (a) shows how the test accuracy drops and saturates as $\sigma$ is increased. (c) shows how gradients start to show misalignment as the scale is increased. (b) shows a similar misalignment trend for hidden layer representations. Note that, in contrast to the extreme memorization phenomenon we observed for sin activation, here we observe a more limited decrease in both generalization performance and alignment. Similar results on CIFAR-100, SVHN and additional plots for CIFAR-10 with all the loss functions discussed in Section 3.2 can be found in the appendix.

**Representation Alignment** Since gradients are the sole contributor to changes in the the weights of the net, they play a crucial part in capturing generalization performance. However, calculating the gradient for every example in the batch can incur a significant compute and memory overhead. Fortunately, the gradients for $W_2$ are a functions of the intermediate representations $r_i = \phi(W_1 x_i)$. Considering example representations instead of example gradients has a practical advantage that representations can be obtained for free with the forward pass. Also, representation alignment, defined below as $\Omega^r$, at any training step, accounts for the cumulative changes made by the gradients since the beginning of the training whereas gradient alignment only accounts for the current step. We show a comparison with gradient alignment in Figure 2. For completeness, we provide plots for both gradient and representation alignment for all the experiments where its useful to do so. We formulate representation alignment in the same way we formulated gradient alignment below.

$$\Omega^r_{in-class} := \frac{1}{k} \sum_{c=1}^{k} \Omega^r_c \qquad\qquad \Omega^r_c := \frac{n_c \sum_{i \neq j} \langle r_i, r_j \rangle \mathbb{1}[y_i = y_j = c]}{(n_c - 1)(\sum_i^n \|r_i\| \mathbb{1}[y_i = c])^2}$$

## 4    WHY SHOULD THE SCALING AFFECT HOMOGENEOUS ACTIVATIONS ?

For sin activations, extreme memorization phenomenon may be explainable through the lens of random Fourier features and kernel machines, which suggests that large initialization leads to very poorly aligned examples. In this Section, we investigate what happens when we use more typical

activations such as ReLU. We find that even for ReLU, increasing the scale of the initialization leads to a drop in generalization performance, and a similar downward movement in alignment as the initialization scale increases (see Figure 4). This might feel counter-intuitive as the ReLU activation is positive-homogeneous, which means that scale factors from the input can be pulled out of the function altogether and it only changes the scale of the output. However, this does not take into account the effect of the loss function $\ell$, which is typically *not* homogeneous. We study 3 commonly used loss functions, namely softmax cross-entropy, multi-class hinge loss and squared loss, and show their effect on gradients when weights are close to their initialization. The result we present for ReLU holds for linear activation too. Even though with linear activations we don't expect perfect training accuracy, we do see the same trend in alignment measures and the drops in generalization performance that goes with it. Due to space constraints, we refer the reader to appendix Section F for the plots.

As shown in Figure 2, increasing the scale of initialization also leads to the scale of the gradients being much smaller than the scale of the parameters at initialization (Chizat & Bach, 2018; Woodworth et al., 2020). Thus if it was high enough in the beginning, SGD should not be able to *correct* the scale of the weights during the course of the training.

**Softmax cross entropy** Typically, the softmax layer consists of a weight vector $s_i$ for every class, which is used to compute the logits $z_i$. These logits then are used to compute the probability $p_i$ for each class using the softmax function $g : \mathbb{R}^k \to \mathbb{R}^k$:

$$p_i = g_i(z) = \frac{e^{z_i/T}}{\sum_{j=1}^k e^{z_j/T}} \text{ for } i = 1, \ldots, k \text{ and } z = (z_1, \ldots, z_k) \in \mathbb{R}^k \tag{7}$$

Assuming $T$ is 1, which is typically the case, the derivative of the loss with respect to $z_i$ is $\frac{d}{dz_i}\ell(g(z), y) = p_i - y_i$ where $\ell$ is the negative log-likelihood and $g(z) = (p_1, \ldots, p_n)$ is the Softmax function. Let us consider the limiting behavior of this gradient when we increase the scale of the network, which causes the $z$ values to become arbitrarily high. In this case, all the $p_i$ except the one corresponding to the largest $z$ value become zero, so that the gradient is $0$ if the prediction is correct, and otherwise is $-1$ in the coordinate of the correct class and $1$ in the coordinate of the predicted class. Contrasting this with the case where the scale of the network is arbitrarily close to 0, the gradient in the coordinate of the correct class will be $1/k - 1.0$ and $1/k$ in the incorrect class coordinates, so that all the gradients are the same and the alignment is 1, which is the maximum possible alignment. Therefore the gradients with respect to the logits will on average be more orthogonal in the former case. Since the gradients for parameters will be multiplied by gradient with respect to the logits due to the chain rule, they will be more orthogonal as well. We corroborate this intuition with empirical evidence as shown in Figure 4.

In practice, since the initialization scheme is chosen carefully, weight scaling is less of a concern in the beginning, but it can become an issue during the course of the training if the magnitude of the weights starts to increase. In either scenario, one simple strategy to counteract the effect of scaling of the net is to increase the temperature term $T$ with it such that magnitude of the input to the Softmax can stay the same and consequently there will be no relative change to alignment in the gradients coming from the loss function. Moreover, this observation also brings some clarity into why tuning hyper-parameters that affects the scale of the network is sometimes helpful in practice, either by explicitly tuning the temperature term or appling weight decay which favors parameters of low norm and implicitly controls the scale of the network throughout the training run.

The arguments made for Softmax can also be adapted to Sigmoid for binary classification. Moreover, since Sigmoid is also used occasionally as an activation function, it is valuable to see how it behaves with changes in scale of initialization in that capacity. We do in fact observe similar degradation in generalization performance, although in this case, there is an extra complication that increasing the scale of the input to Sigmoid also affects the training accuracy since gradients for the hidden layer starts to saturate beyond a certain scale. More details on this can be found in the appendix.

**Hinge loss** is defined by:

$$\ell(z, y) = \sum_{i \neq y} \max(0, \Delta + z_i - z_y) \tag{8}$$

where $\Delta$ is the target margin. In practice $\Delta$ is typically set to 1.0. However, if the network outputs are scaled by a factor of $\alpha$, this will have the same effect as scaling the margin to be $\frac{\Delta}{\alpha}$ and then

scaling the loss by $\alpha$: $\sum_{i \neq y} \max(0, \Delta + \alpha z_i - \alpha z_y) = \alpha \sum_{i \neq y} \max(0, \Delta/\alpha + z_i - z_y)$. With this in mind, let us calculate the gradient:

$$\frac{d\ell}{dz_i} = \begin{cases} \mathbb{1}(\Delta + z_i - z_y > 0) & i \neq y \\ -\sum_{i \neq y} \mathbb{1}(\Delta + z_i - z_y > 0) & i = y \end{cases} \quad (9)$$

It is instructive to take a look at what happens when the effective margin is arbitrarily close to zero. At initialization, we can treat each $\mathbb{1}(z_i - z_y > 0)$ as independently 0 or 1 uniformly at random, so we can expect half of the gradient coordinates for incorrect classes to be 1. On the other extreme, if the effective margin becomes large $\mathbb{1}(\Delta + z_i - z_y > 0)$ will always be 1, and the gradients for all incorrect classes will be 1. Again, the latter case will lead to the maximum alignment value of 1, so that the gradients more aligned across examples.

In this case, misalignment can be fixed by scaling the margin $\Delta$ with the scale of the network. Intuitively, we want to change the loss function such that the scale factor can be pulled out of the loss entirely so that scaling of the loss by a constant doesn't change the minimizer.

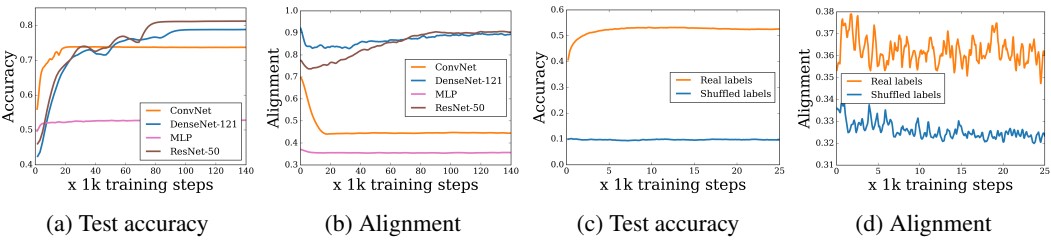

(a) Test accuracy       (b) Alignment       (c) Test accuracy       (d) Alignment

Figure 5: Plots (a) and (b) shows how representation alignment increases with generalization performance as the architecture is improved from 2-layer MLP with ReLU activation to a ConvNet architecture (exact details in the appendix) on CIFAR-10 dataset. We see further increase when even bigger and widely used architectures like ResNet-50 and DenseNet-121 are employed on the same task. Note that we keep all the hyperparameters same across architectures in this experiment. Plots (c) and (d) elucidates the drop in representation alignment when the labels are shuffled in the case of 2-layer MLP.

**Squared loss** is defined as $\ell(z, y) = \frac{1}{2}(z - y)^2$ and $\frac{d\ell}{dz} = z - y$, where $y$ is a one hot vector with 1 in the coordinate of the correct class. In the extreme case where scale of the net is close to 0, $z$ will also be close to zero so the $y$ term will dominate in gradients for all the examples. On the other hand, when the scale is high, the $z$ term dominates. Again, since $y$ is a constant in our training and $z$ will essentially be a random vector at initialization, we can expect the gradients across examples to be more aligned when the scale of the network is lower. Similar to the argument presented for hinge loss, the effect of scaling on generalization performance in this case can be fixed by scaling the one-hot vector $y$ appropriately with it so that the scale factor can be pulled out of the loss function.

## 5   IS ALIGNMENT RELEVANT MORE BROADLY?

In this Section, we explore whether the alignment metric is useful in capturing generalization performance more generally. More specifically, is alignment relevant when we make changes to architecture or data distribution rather than the initialization scheme? We provide empirical evidence which suggests an optimistic answer.

Introduction of new architecture changes has been a very successful recipe in advancing performance of deep learning models. In the task of image recognition, addition of convolutional layers and pooling layers (Lecun et al., 1998) and, more recently, residual layers (He et al., 2015b; 2016) have caused significant jumps in generalization performance. Moreover, several theoretical studies show how convolution and pooling operations can significantly affect the implicit bias of SGD, in favor of better generalization in the image domain (Cohen & Shashua, 2016; Gunasekar et al., 2018). In Figure 5, we investigate the architectural change of extending our standard 2-layer MLP with preceding convolutional layers. Unsurprisingly, we observe substantial improvement in generalization performance. Moreover, the plots show that the addition of convolutional layers leads the last layer representations to be significantly more aligned, suggesting that these architecture changes cause the

net to discard irrelevant variations in the input more effectively across examples and ultimately leads to better generalization. Finally, we experiment with popular large scale image recognition models like ResNet-50 and DenseNet-121 (Huang et al., 2017) and also observe a similar trend.

Another way to impact generalization performance is to shuffle the labels in the training set (Zhang et al., 2016; Arpit et al., 2017). If we completely shuffle the labels in the dataset, we don't expect the model to generalize on the test set at all, i.e., random chance performance. Fig 5 shows that shuffling the labels also leads to a drop in representation alignment.

## 6    CONCLUSION AND FUTURE WORK

In this work, investigated how increasing the scale of initialization can degrade the generalization ability of neural nets. We observed an extreme case of this phenomenon in the case of sin activation, making it particularly interesting given a recent rise in the use of sin activation in practical setting (Sitzmann et al., 2020; Tancik et al., 2020), since our work shows how sensitive the generalization performance can be to scale of initialization in those case. This phenomena is also quite conspicuous even with more popular activations like ReLU and Sigmoid. In the case of ReLU, we argue that the drop in generalization performance can be attributed to the loss function since the rest of the net is unaffected by scaling due to homogeneity. We complement these observations by defining an alignment measure that correlates empirically well with generalization in a variety of settings, including the inductive bias introduced by architectural changes like convolution layers, indicating its broader importance.

Our formulation of alignment measure suggests some intriguing avenues for future research. For example, as shown in Figure 1, even though our experiments suggest that low scale of initialization leads to increased representational alignment, there seems to be a sweet spot below which its affect on generalization ability no longer holds true. Exploring generalization behavior in this case of ultra-low scale of initialization is also an interesting direction of future research.

## 7    ACKNOWLEDGEMENTS

We are grateful to Eugene Ie, Walid Krichene and Jascha Sohl-dickstein for reading earlier drafts of this paper and providing valuable feedback.

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

## A    ORGANIZATION OF THE APPENDIX

- We describe the training procedure and datasets used throughout the paper in detail in appendix C.
- In appendix D, we reproduce the extreme memorization phenomenon from Figure 2 on CIFAR-100 and SVHN.
- In Section 3.2, Fig 4 shows how changing the scale of activation leads to a drop in generalization performance in the case of ReLU activation and softmax cross-entropy loss. In appendix E, we include results with multi-class hinge and squared losses.
- We include results when using linear activation with softmax cross-entropy loss in appendix F.
- In appendix G, we discuss how Sigmoid function, when used as activation, responds to scaling of initialization.
- Appendix H includes details on the exact architecture and hyperparameters used in Section 5.

## B    PROOF OF THEOREM 1

In this section we provide the missing proof of Theorem 1, restated below:

**Theorem 1.** *Suppose each entry of $W_1$ is initialized via a Gaussian with mean 0 and variance $\sigma^2$. Then for any $x$ and $x'$, we have*

$$\left| \mathop{\mathbb{E}}_{W_1}\left[ \langle \phi(W_1 x), \phi(W_1 x') \rangle \right] \right| \le h \exp\left( -\frac{\sigma^2 \|x - x'\|^2}{2} \right)$$

*Proof.* Since each individual row of $W_1$ is independent, it suffices to prove the statement for $h = 1$. If $x = 0$ the statement is trivially true, so suppose $x \ne 0$. Let $c = \frac{\langle x', x \rangle}{\|x\|^2}$ and let $\Delta = x' - cx$. Notice that $\langle \Delta, x \rangle = 0$ and $\|\Delta\| \le \|x - x'\|$. We also have

$$W_1 x' = c W_1 x + W_1 \Delta$$

Notice that $W_1 x$ is normally distributed with mean 0 and variance $\sigma^2 \|x\|^2$. Further, $W_1 \Delta$ is normally distributed with mean 0 and variance $\sigma^2 \|\Delta\|^2$. Let $A$ be a mean 0 random variable with variance $\sigma^2 \|x\|^2$ and $B$ be a mean 0 random variable with variance $\sigma^2 \|\Delta\|^2$. Notice that since $\langle \Delta, x \rangle = 0$, the joint distribution $(W_1 x, W_1 x')$ is the same as that of $(A, cA + B)$. Therefore we have:

$$\begin{aligned}
\left| \mathop{\mathbb{E}}_{W_1}\left[ \langle \phi(W_1 x), \phi(W_1 x') \rangle \right] \right| &= \left| \mathop{\mathbb{E}}_{A,B}\left[ \sin(A)\sin(cA + B) \right] \right| \\
&= \left| \mathbb{E}[\sin(A)\sin(cA)\cos(B) + \sin(A)\cos(cA)\sin(B)] \right| \\
&= \left| \mathbb{E}[\sin(A)\sin(cA)\cos(B)] \right| \\
&= \left| \mathbb{E}[\sin(A)\sin(cA)]\, \mathbb{E}[\cos(B)] \right| \\
&\le \left| \mathbb{E}[\cos(B)] \right| \le \exp\left( -\frac{\sigma^2 \|\Delta\|^2}{2} \right)
\end{aligned}$$

$\square$

## C    DESCRIPTION OF THE TRAINING PROCEDURE AND DATASETS

We use the Tensorflow framework for conducting our empirical study and all of our code is included as part of supplementary material. In every experiment, we train using SGD, without momentum, with a constant learning rate of 0.01 and batch size of 256. We employ a p100 single-instance GPU for each training run. For most of the experiments, the model is trained until it obtains perfect accuracy on the training set, with only a few exceptions which are either unavoidable or requires extravagant training iterations. For example, in the experiments involving linear activation, since none of the datasets we use are completely linearly separable, we do not expect the net to get 100% accuracy on

the training set. Another interesting case is the Sigmoid activation, for which the gradients starts to saturate as the scale of the input to the Sigmoid function increases. Thus, we stop the training at a point when at least one of the model in the study achieves perfect accuracy on the training set.

In our 2-layer MLP model, in almost all cases we use 1024 units for the hidden layer with exceptions of 1) experiments with Sigmoid activation and 2) ReLU activation with squared loss. In both of these cases, we increase the number of hidden units to 2048 in order to increase their training speed. Number of units for the softmax layer depends on the number of output classes, which is 10 for CIFAR-10 / SVHN, and 100 for CIFAR-100. The details of the ConvNet architecture are included in appendix H. For any layer that doesn't involve changing the initialization scale, for instance the top layer in all our models, defaults to using Glorot uniform initializer (Glorot & Bengio, 2010). For experiments corresponding to Sections 3 and 3.2, we refrain from employing bias variables in order to match the setup exactly. For experiments in Section 5, all biases are initialized to zero.

We employ 3 image classification datasets each having 32x32 pixels color image as input. CIFAR-10 dataset (Krizhevsky, 2009) consists of 60000 images with 10 classes. Classes are balanced with 6000 images per class. Training set consists of 50000 images and 10000 test images. CIFAR-100 (Krizhevsky et al.) is very similar to CIFAR-10 except that it has 100 classes with 600 images per class. Finally, The Street View House Numbers (SVHN) Dataset (Netzer et al., 2011) has images of digits from house numbers obtained from Google Street View with a total of 10 classes. Training set contains 73257 images and 26032 test images.

## D  SIN ACTIVATION

Figure 2 shows that increasing the scale of initialization for hidden layer weights $W_1$ in a 2-layer MLP model leads to extreme memorization on CIFAR-10 dataset. Keeping everything else the same, we reproduce the same phenomenon on two other datasets, namely, CIFAR-100 and SVHN respectively.

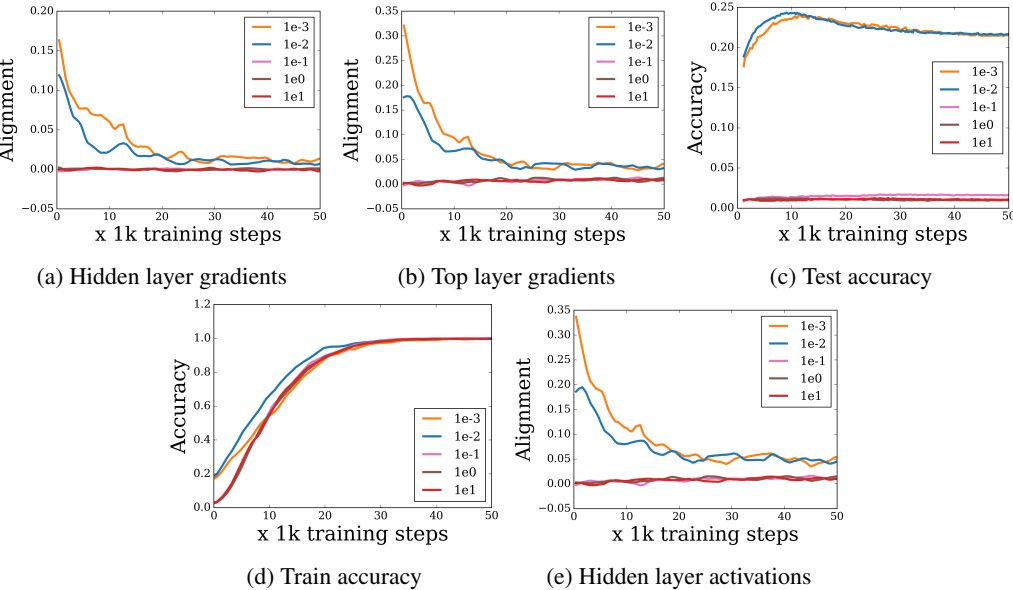

Figure 6: Results when using sin activation function on CIFAR-100 dataset.

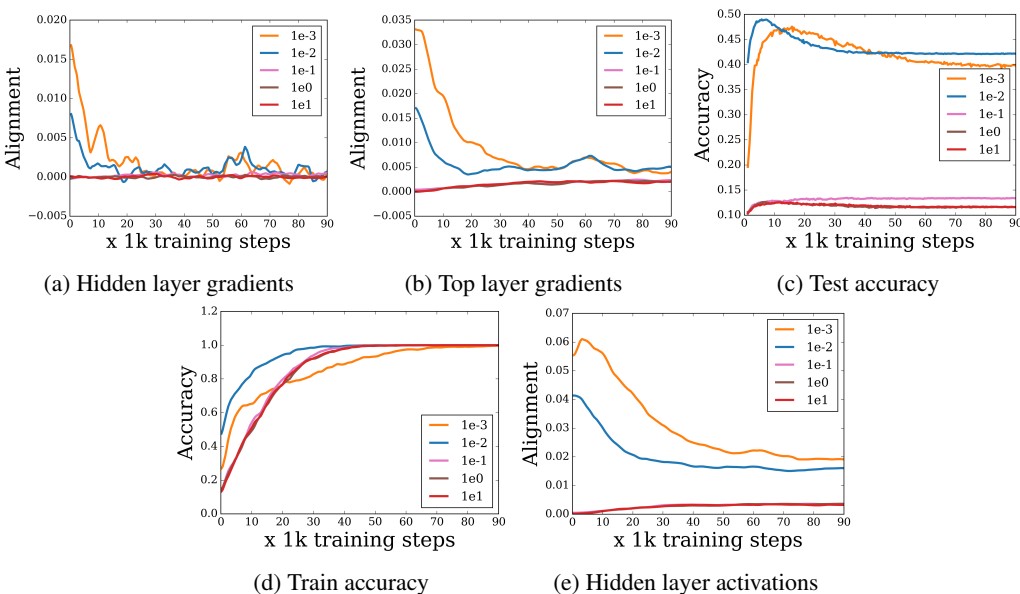

Figure 7: Results when using sin activation function on SVHN dataset.

## D.1 VARYING LEARNING RATES

Figure 2 also illustrates the lazy training phenomenon, in which gradients for the first layer are smaller compared to the weights of the net. Thus, one might worry that the lack of generalization behavior is due to some scaling mismatch between the weights and the gradients - essentially due to a poor learning rate choice - rather than some modified implicit bias due to modified scaling. To investigate this possibility, we conduct the following study where we selectively alter the learning rate for the first layer. As shown in the figure below, we observe extreme memorization phenomenon when the scale of the net is high across a wide range of learning rates. Note that increasing the learning rate after a certain point induces the net to stop learning altogether i.e. it no longer achieves perfect *training* accuracy.

As a special case, we also experiment with frozen first layer weights. Even though, in this case, the net fails to achieve perfect training accuracy, we can still see the extreme memorization phenomenon when the scale of initialization is high enough.

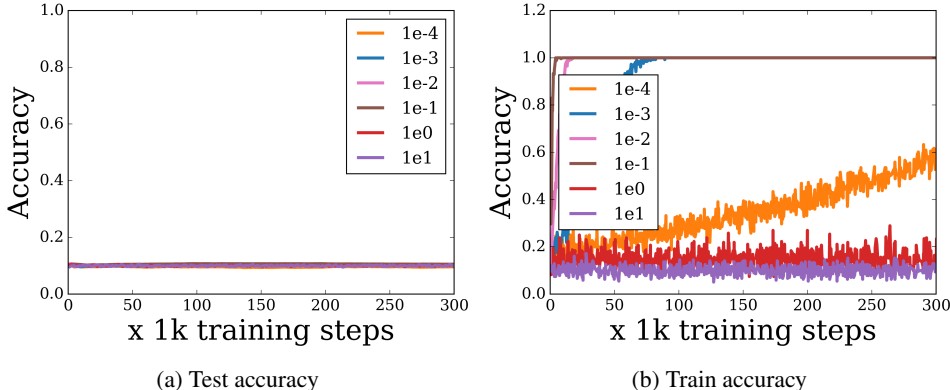

Figure 8: Results when using sin activation function on CIFAR-10 dataset with scale of initialization set to 1.0 and varying learning rates for the first layer as shown in the plot.

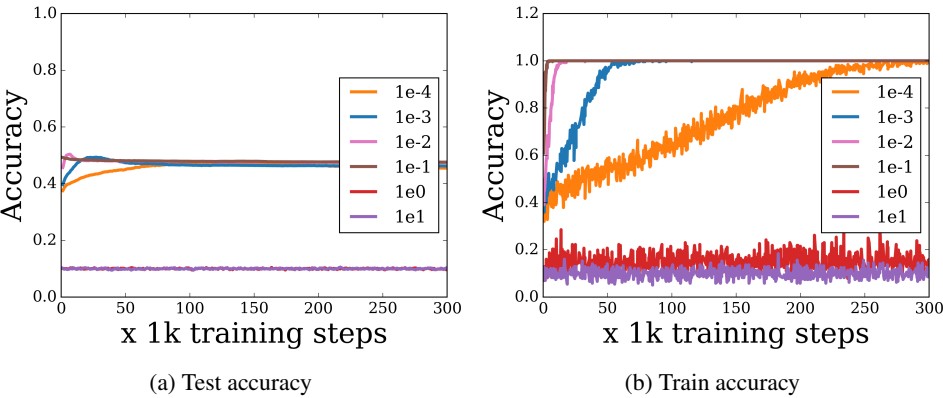

(a) Test accuracy

(b) Train accuracy

Figure 9: Results when using sin activation function on CIFAR-10 dataset with scale of initialization set to 1e-2 and varying learning rates for the first layer as shown in the plot.

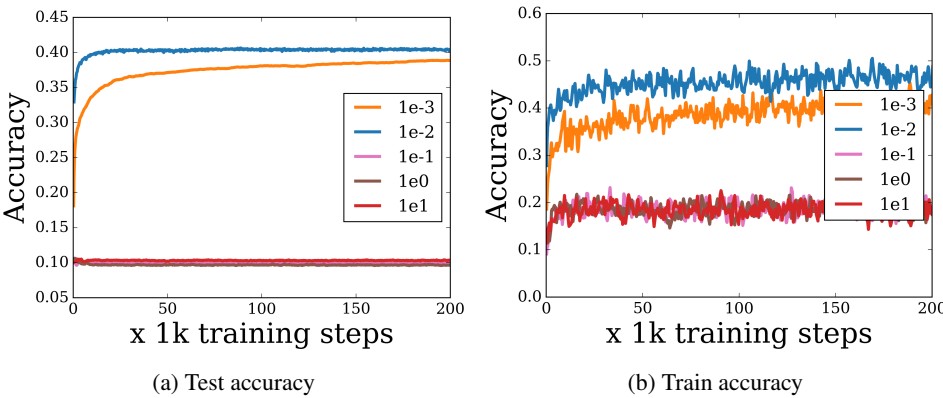

(a) Test accuracy

(b) Train accuracy

Figure 10: Results when using sin activation function on CIFAR-10 dataset when the first layer weights are frozen and varying scale of initialization as shown in the plot.

## D.2 DOES DEPTH MATTER?

We also experiment with increasing the depth of the MLP from 2-layers to 4-layers. We find the extremem memorization is present even in this case. We also see a similar decrease in generalization performance in the case of ReLU.

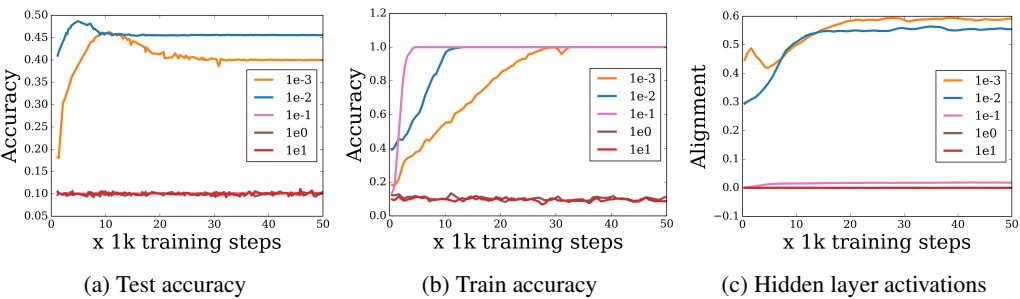

(a) Test accuracy

(b) Train accuracy

(c) Hidden layer activations

Figure 11: Results when using sin activation function on CIFAR-10 dataset with 4-layer MLP.

# E ReLU ACTIVATION

## E.1 SOFTMAX CROSS ENTROPY

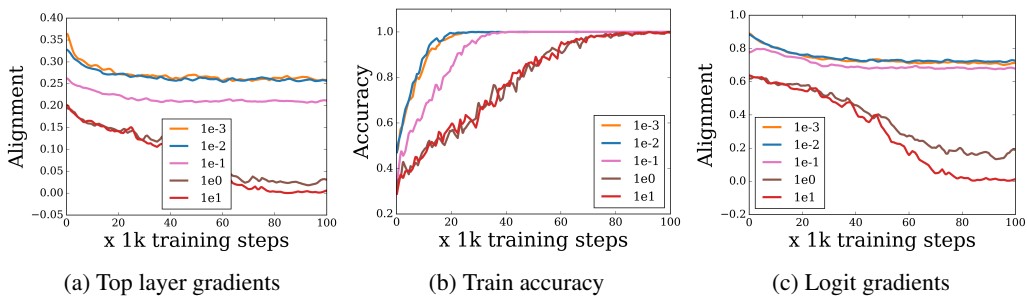

(a) Top layer gradients      (b) Train accuracy      (c) Logit gradients

Figure 12: Additional plots when using ReLU activation function with softmax cross-entropy on CIFAR-10 dataset.

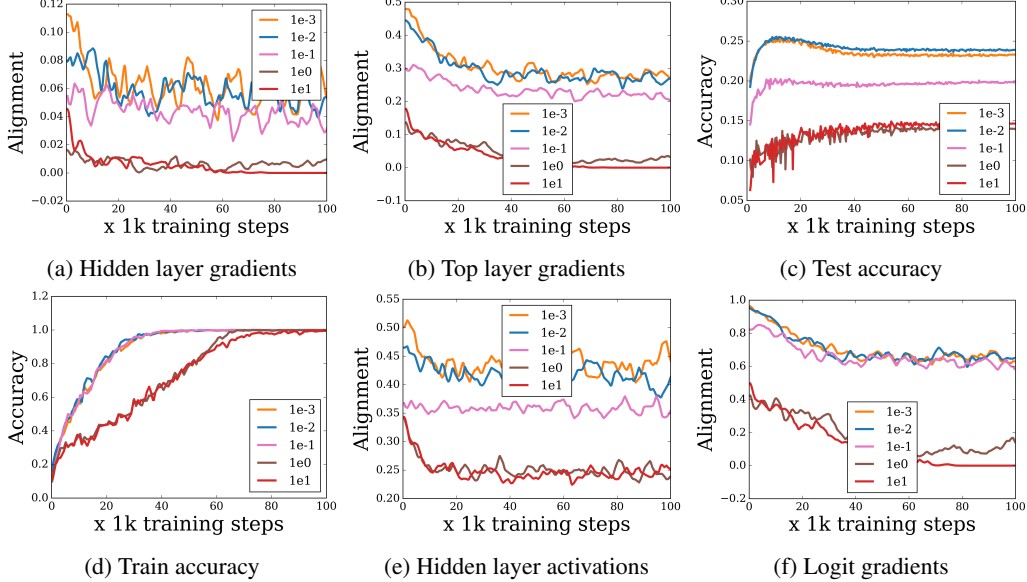

(a) Hidden layer gradients      (b) Top layer gradients      (c) Test accuracy

(d) Train accuracy      (e) Hidden layer activations      (f) Logit gradients

Figure 13: Results when using ReLU activation function with softmax cross-entropy on CIFAR-100 dataset.

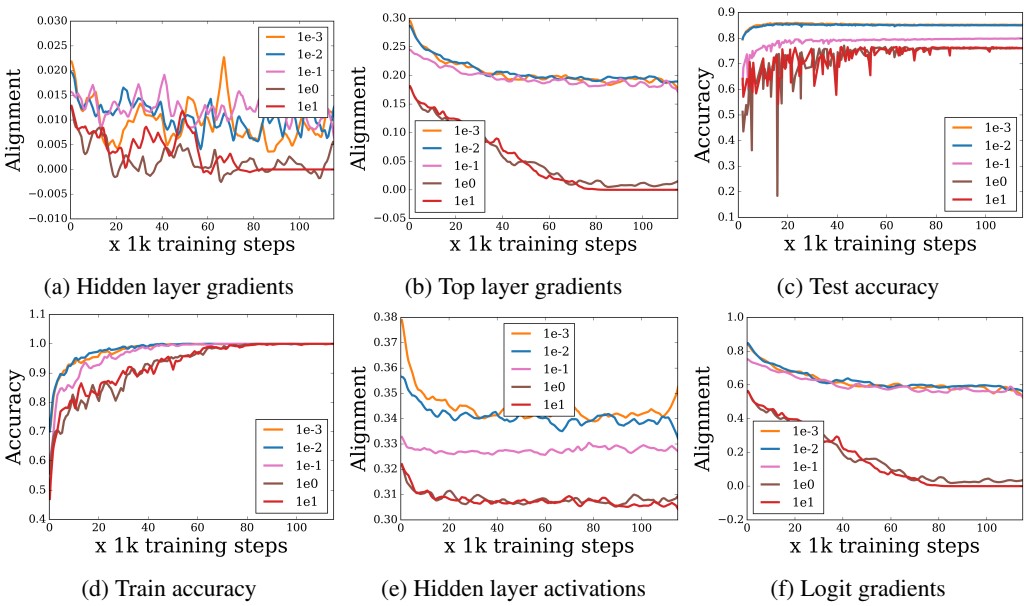

Figure 14: Results when using ReLU activation function with softmax cross-entropy on SVHN dataset.

## E.2 HINGE LOSS

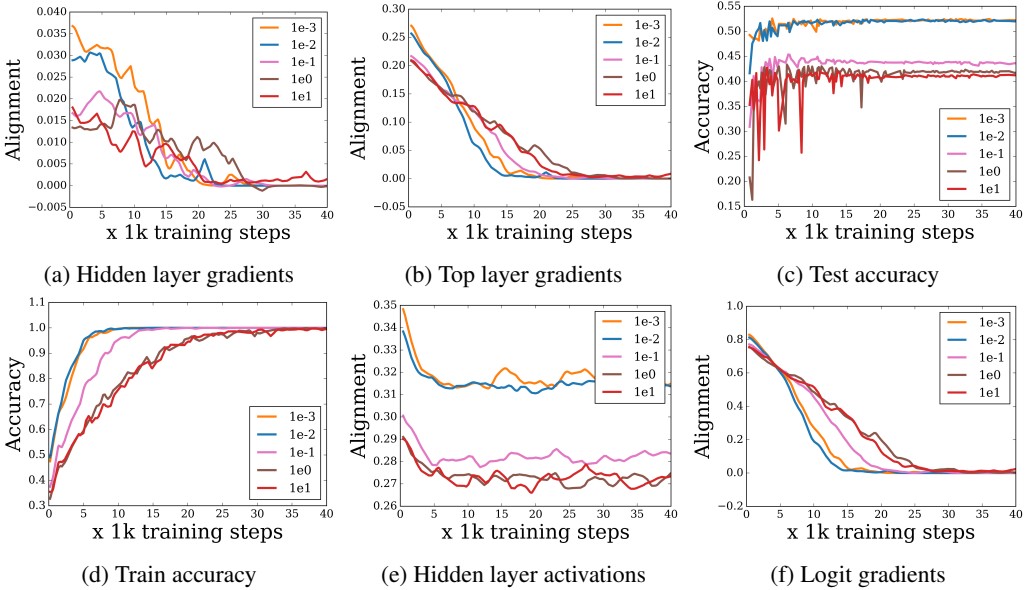

Figure 15: Results when using ReLU activation function with hinge loss on CIFAR-10 dataset.

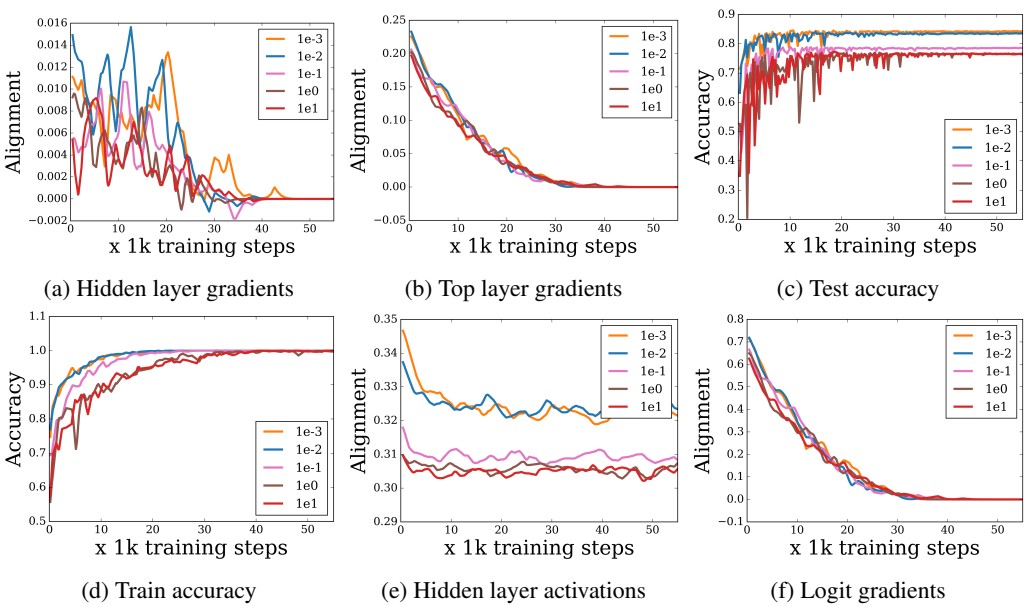

Figure 16: Results when using ReLU activation function with hinge loss on SVHN dataset.

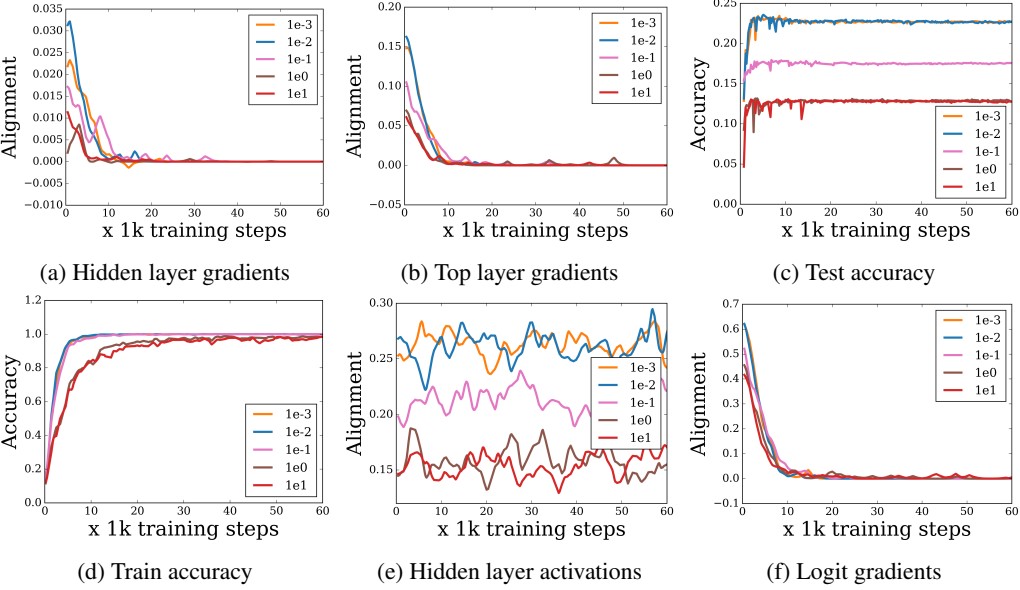

Figure 17: Results when using ReLU activation function with hinge loss on CIFAR-100 dataset.

### E.3 DOES DEPTH MATTER?

Similar to sin activation, we see a decrease in generalization performance even in the case of ReLU, despite increasing the depth.

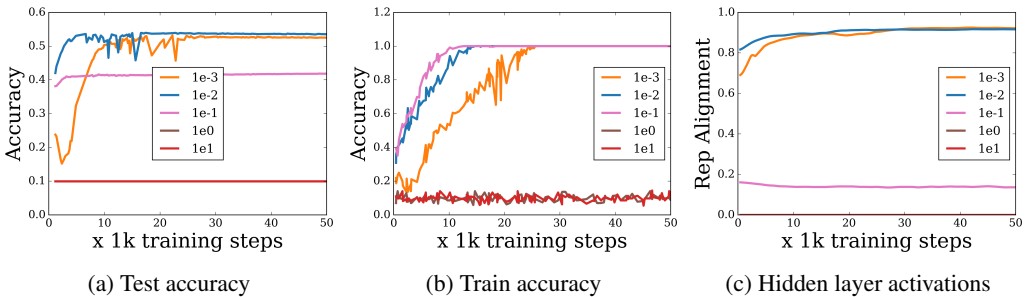

(a) Test accuracy       (b) Train accuracy       (c) Hidden layer activations

Figure 18: Results when using ReLU activation function on CIFAR-10 dataset with 4-layer MLP.

### E.4 SQUARED LOSS

Note that when employing squared loss, we increase the number of hidden units from the usual 1024 units to 2048 in order to compensate for very low training speed. Also, we observed that increasing the scale of initialization for $W_1$ beyond a certain scale leads to divergence in training after a few iterations. Thus, we recover the phenomenon of interest with much less aggresive increase in scale of initialization i.e. we double the standard deviation instead of increasing it by ten times as done in other experiments.

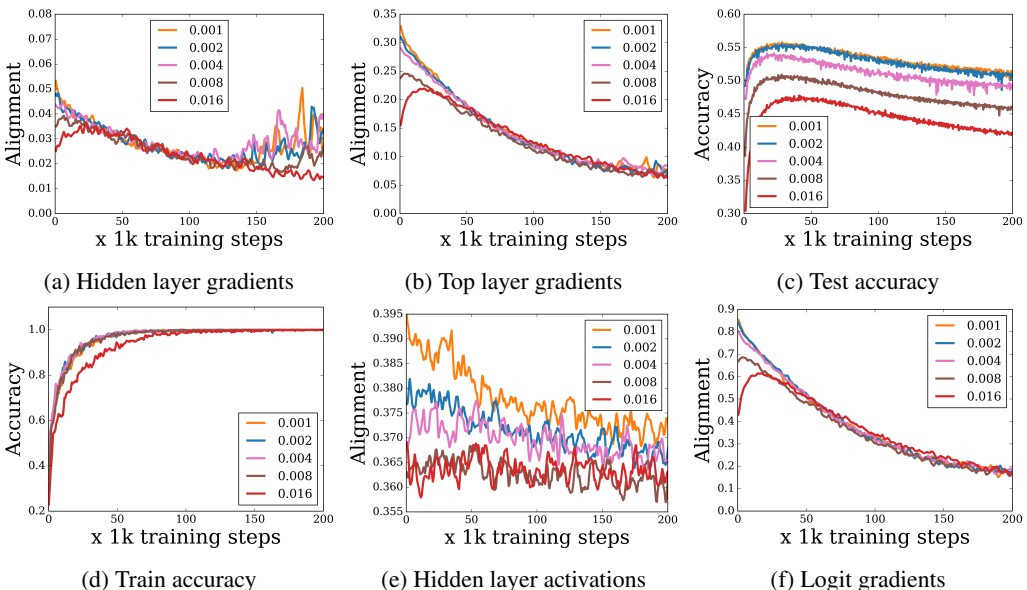

(a) Hidden layer gradients       (b) Top layer gradients       (c) Test accuracy

(d) Train accuracy       (e) Hidden layer activations       (f) Logit gradients

Figure 19: Results when using ReLU activation function with squared loss on CIFAR-10 dataset.

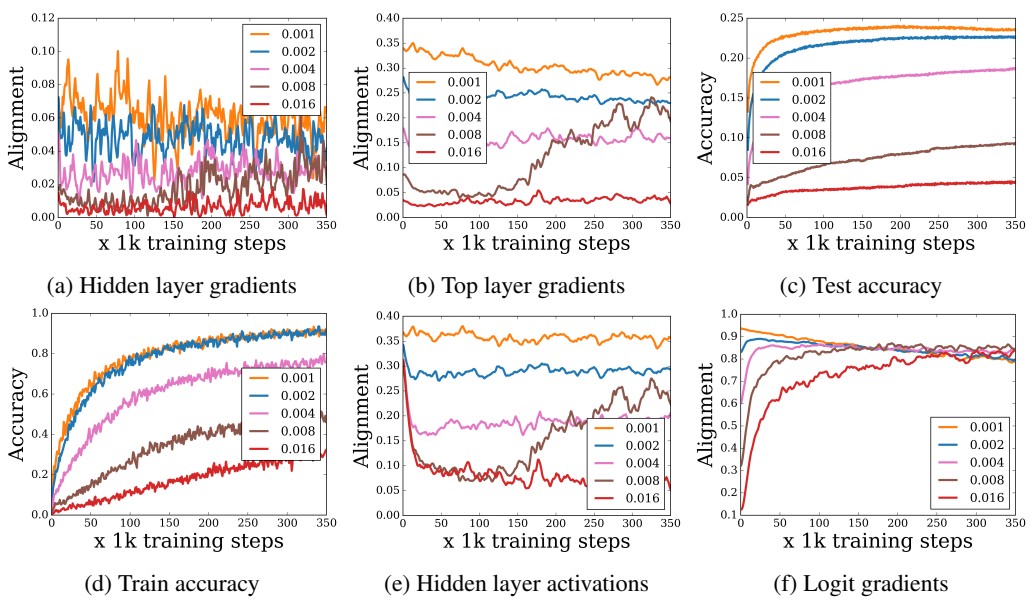

(a) Hidden layer gradients     (b) Top layer gradients     (c) Test accuracy

(d) Train accuracy     (e) Hidden layer activations     (f) Logit gradients

Figure 20: Results when using ReLU activation function with squared loss on CIFAR-100 dataset.

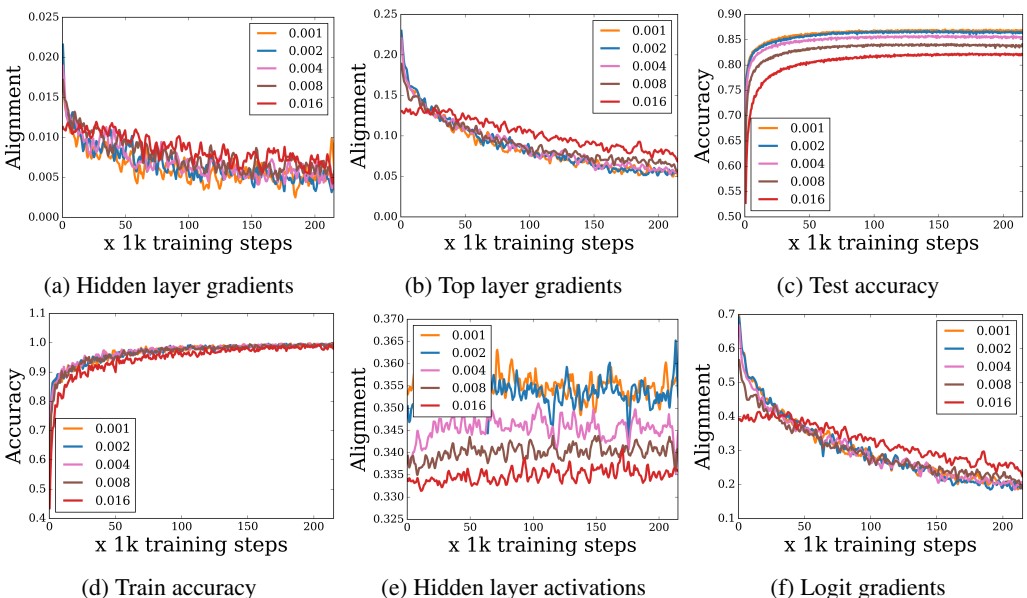

(a) Hidden layer gradients     (b) Top layer gradients     (c) Test accuracy

(d) Train accuracy     (e) Hidden layer activations     (f) Logit gradients

Figure 21: Results when using ReLU activation function with squared loss on SVHN dataset.

## F LINEAR ACTIVATION

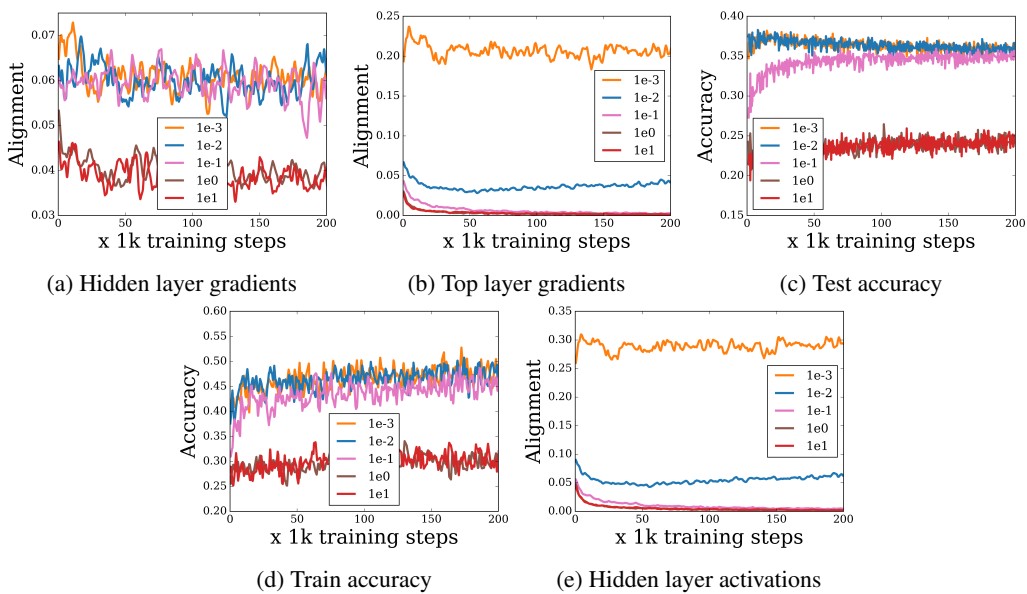

(a) Hidden layer gradients      (b) Top layer gradients      (c) Test accuracy

(d) Train accuracy      (e) Hidden layer activations

Figure 22: Results when using linear activation function with softmax cross entropy loss on CIFAR-10 dataset.

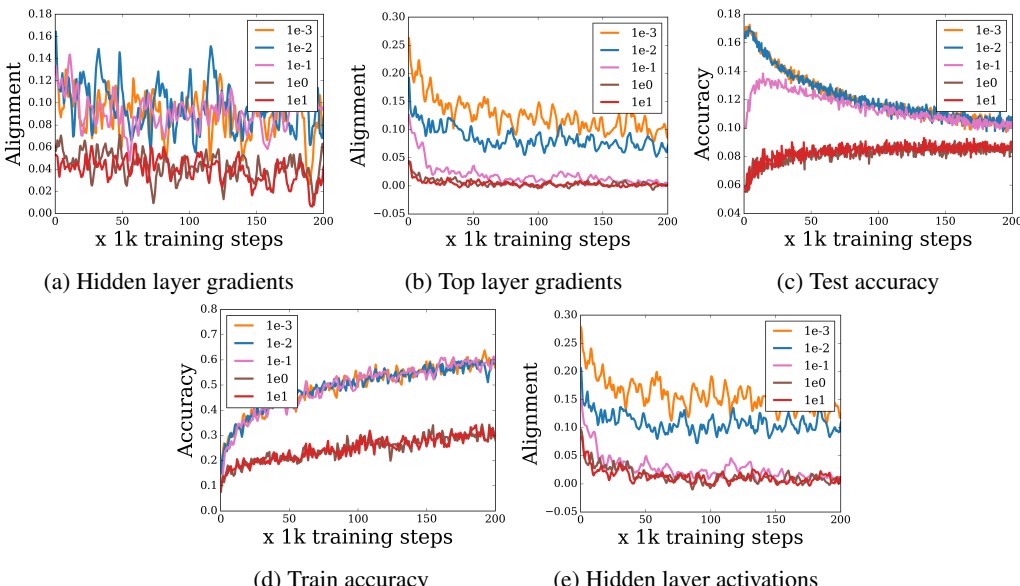

(a) Hidden layer gradients      (b) Top layer gradients      (c) Test accuracy

(d) Train accuracy      (e) Hidden layer activations

Figure 23: Results when using linear activation function with softmax cross entropy loss on CIFAR-100 dataset.

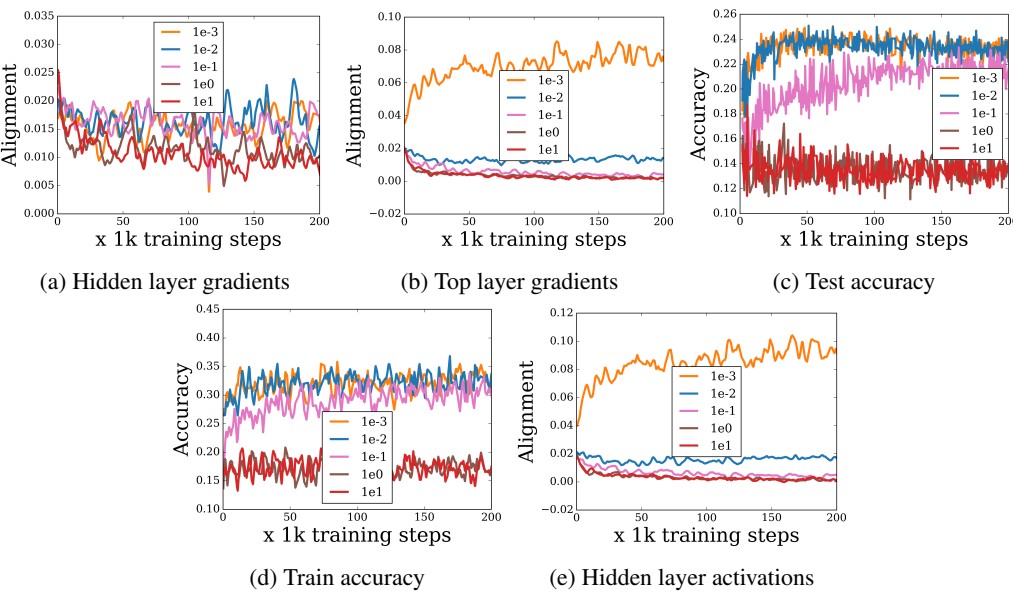

Figure 24: Results when using linear activation function with softmax cross entropy loss on SVHN dataset.

# G   SIGMOID ACTIVATION

Note that when employing Sigmoid activation, after a certain scale the hidden layer gradients start to vanish. We try to compensate for this by increasing the number of hidden units to 2048.

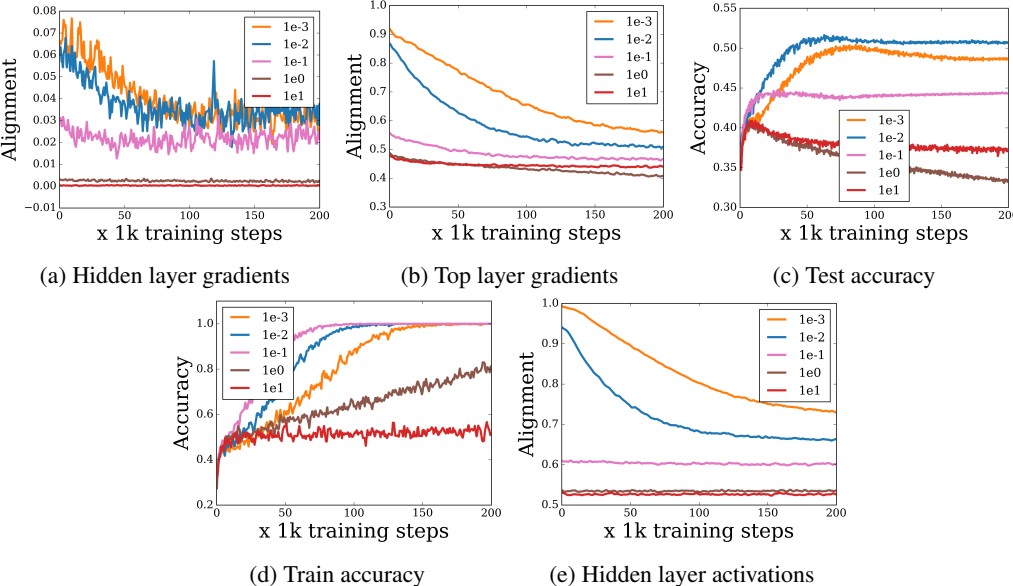

Figure 25: Results when using Sigmoid activation function on CIFAR10 dataset.

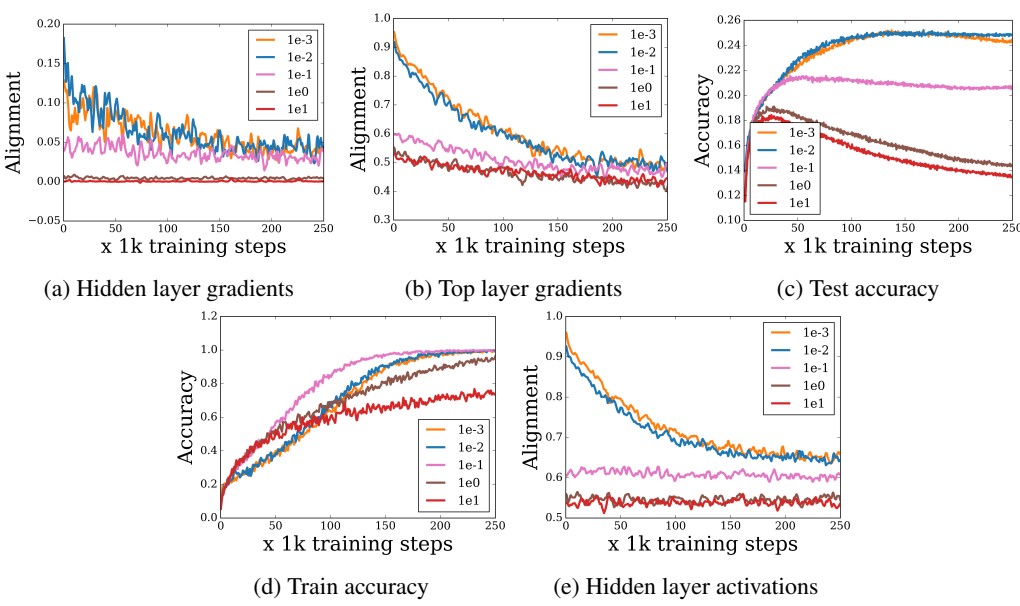

Figure 26: Results when using Sigmoid activation function on CIFAR100 dataset.

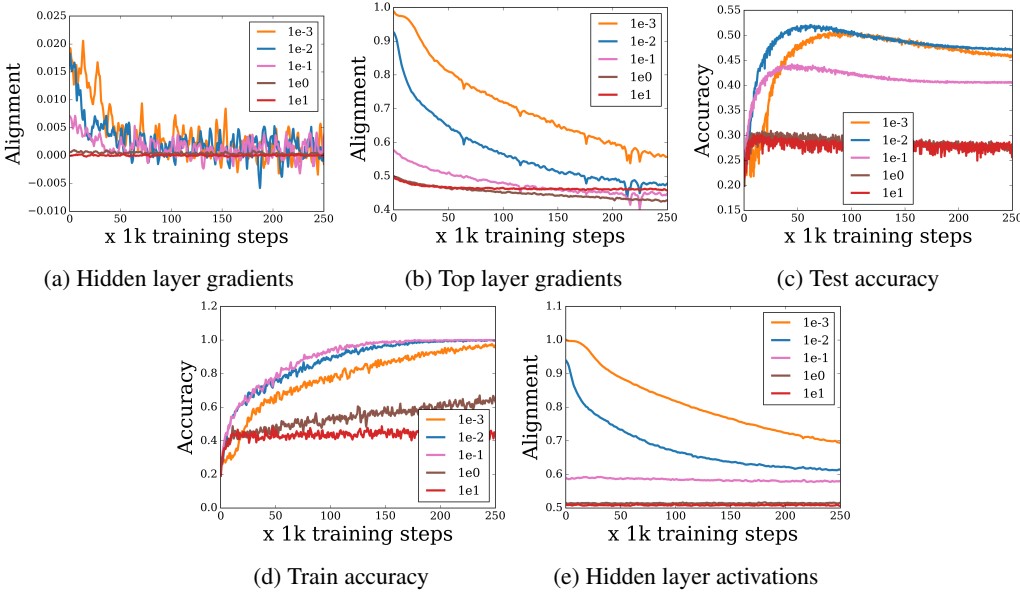

Figure 27: Results when using Sigmoid activation function on SVHN dataset.

# H CONVNET ARCHITECTURE

Our goal is to recover the phenomenon that convolution and pooling operation leads to more aligned representations. In order to do this, we construct a simple ConvNet architecture. We start with two consecutive convolution and max pool operations, followed by a fully-connected and softmax layers. Note that the last two layers fully connected and softmax are operationally the same as our 2-layer MLP.

We keep all the other hyper parameters the same between training runs for all the architectures. All the models are trained with SGD without momentum with learning rate set to 0.01 and batch size to 256.

- Convolution layer 1: 32 filters with 5x5 kernel size followed by ReLU activation.
- Max pooling layer 1: pool size 3x3 with stride of 2x2
- Convolution layer 2: 64 filters with 5x5 kernel size followed by ReLU activation.
- Max pooling layer 2: pool size 3x3 with stride of 2x2
- Fully connected layer with 1024 hidden units followed by ReLU activation.
- Softmax layer with 10 output logits.

