# OpenReview forum: "Extreme Memorization via Scale of Initialization"
_ICLR.cc/2021/Conference — ICLR 2021 Poster_

### Official Review · AnonReviewer3 · 2020-10-26
**Paper is interesting, but I am not sure about the experiment setup**

**Rating:** 9
**Confidence:** 4

**Review:**

## Overview

The paper studies how the generalization of the neural network trained with SGD is affected by the scale of the random initialization. Specifically, that when the scale of the initialization is big, the network overfits to the training set with bad performance on the test set.
The paper also provides a hypothesis why does it happen -- that the graidents of difference examples are orthogonal in the "bad" mode and proposes a measure called "alignment" to predict the generalization reghime of the network.

The problem, tackled in paper is interesting and the paper itself is thought-provoking.


The central model studied is 2 layer fully-connected neural network with {sin, ReLU} activation, where the 1st layer is initialized with variable (studies) scale and the 2nd - with Xavier init.
Both layers are without biases. The model is optimized with SGD w/o momentum and constant learning rate and the loss function is one of the classification losses: CE, hinge loss.

## Strong points

The paper formulates hypothesis and verifies it via series of controlled experiments. Besides "toy" 2-layer model, the similar results are get with more powerful architectures, like CNN, DenseNet and so on on the set of middle-sized datasets like SVHN and CIFAR.
The paper clearly states its place among related works and proposes a useful metric for diagnosing model training.

## Questions (and possible weak points


1. Figure 2 shows that the norm(grad)/norm(weight) decreases significantly with the scaling-up the initalization. This is hypothesized to be one of the problems with large scale initialization: the weights do not go far from the original (random) values. If this is the case and the problem, may be scaling up the learning rate with the initalization to keep the norm constant, would help?
Why is this an important question? Because the problem with large init is the bad gradient direction (as hypothesized in paper), then scaling the learning rate would not help. If, otherwise, the problem is the learning rate scale, then the work could be seen as indirect confirmation of the works about large-vs-small learning rate reghimes, e.g.
Lewkowycz et.al.  https://arxiv.org/pdf/2003.02218.pdf
Moreover, may be then we should study not the scale of init, but the ratio (init scale/lr).


2. Following Q1, one also could use different learning rates for different layers depending on the scale of weights and/or activation. Yes, that would significantly complicate the experimental setup, but may lead to the different conclusions. E.g. it is commonly know that deep sigmoid networks are hard to train because of vanishing gradient issue. However, as it was shown in "Revise Saturated Activation Functions" Xu et.al (https://openreview.net/forum?id=D1VDjyJjXF5jEJ1zfE53), one could perfectly fine train such networks. The only thing, which is needed, is the proper rescaling of the learning rate per layer, up to ridiculous values such as 4^11.


3. (minor) why do you don't use bias? Bias-less NNs are less common and give worse results.

4.  (just curious) Where is the actual learning with large scale init take place - in layer 1 or 2? If only one layer is actually trained, can we obtain the same results, when the non-training layer weights are frozen?


5. When examining Figures in appendix (e.g. 17, 20, etc), it looks like that proposed "alignment" measure can be used as alarm -- if it is low, the generalization is low, but the ranking of (test accuracy) and (alignment) is not really aligned. Any comments on that?

Overall, I like the paper, although would also like to have answers to my questions.

## Update after rebuttal

All my concerns have been addressed, including the possible alternative explanation of the experimental results. I strongly recommend the paper to be accepted.

---

> ### Author Response · Authors · 2020-11-17
> **Thank you for your encouraging review! We have added more experiments to answer your questions!**
>
> Thank you so much for your thoughtful review. We have added more experiments in order to fully answer all your questions. We hope that you would consider strongly supporting acceptance of our paper.
>
> (1) and (2):
> That it is a great observation. We fully agree that it is a study worth conducting. To that effect, we updated our paper with 2 experiments in the appendix.
>
> [Appendix Fig 9] - low scale:  Fix the scale of initialization to 1e-2 and vary the learning rate of the first layer (W1 in the paper) \in [1e-4, 1e-3, 1e-2, 1e-1, 1e0, 1e1]
> [Appendix Fig 8]  - high scale:  Fix the scale of initialization to 1.0 and vary the learning rate of the first layer (W1 in the paper) \in [1e-4, 1e-3, 1e-2, 1e-1, 1e0, 1e1]
>
> As we would expect, these experiments provide supporting evidence that the gradients are qualitatively different (misaligned) when the scale of initialization is high and this effect is not tied to the relative scaling of the gradient or the learning rate scale or just due to the fact the weights are almost frozen.
>
> (3): That is a fair question. We did it this way since we wanted to make our setup as simple as possible so that we can analyze it clearly without losing the essence of what we wanted to study. All our conclusions w.r.t scale of initialization also stand when biases are introduced. Also, note that we use standardized architectures (with biases) for all our experiments in Section 5.
>
> (4): Great question! The answer is slightly more nuanced than one would expect. In addition to the experiments we mentioned earlier, we also include results [Appendix Fig 10] for what happens when we set the learning rate for W1 to 0 (freezing the weights). In this case, the net fails to obtain a perfect training accuracy altogether. This is in contrast to when W1 is allowed to train. Even for very high scale of initialization where gradients are much smaller than the weights, the net manages to fully fit the training set. Thus both W1 and W2 are contributing to the learning across all the scales of initialization we tried. But barring this, our conclusion that at a high scale of initialization the net would not be able to learn anything about the test set still stands.
>
> (5) Yes, that is correct. When the alignment is high other factors can also affect the final generalization performance. But when it is lower, we observe that it strongly correlates with how much the net generalizes.

---

### Official Review · AnonReviewer1 · 2020-10-28
**Memorization and Scale Initialization**

**Rating:** 7
**Confidence:** 4

**Review:**

###
Summary:
This paper investigates the role of scale in generalization of neural networks. It shows that the initial scale of a 2-layers MLP, with sinusoid or ReLU activation, can control the memorization behavior of the network, from very little overfitting to complete memorization.
It then proposes an alignment measure which correlates with generalization for different initial scale. It shows that this alignment measure can capture generalization performances for other architecture such as ResNet or DenseNet on the CIFAR-10 dataset.

###
Reasons for score:
Overall, I find the paper to be a bit borderline.
The observation regarding the scale impacting generalization is novel and interesting as I would have assumed that large initial scale would lead to bad optimization rather than a lack of generalization.
However, all the experiments regarding the scale are carried out on a two-layers MLP models and it is not clear to me if similar conclusion would be true for deeper architecture.


###
 Pros:
- interesting observation regarding the impact of the initial scale on generalization
- clearly show the effect in a two-layers MLP with various activation and loss functions
- propose an alignment measure which have some promising correlation with generalization

###
Cons:
- experiments investigating the impact of the initialization scales only for two layers MLPs
- Alignment is not compared with other generalization metrics in section 5.

---

> ### Author Response · Authors · 2020-11-17
> **Thank you for your review! We have added more experiments to address your concerns!**
>
> Thank you for your comments and careful consideration! Below we discuss how we have addressed both of your concerns. We hope that our response alleviates some of your reservations.
>
>
> **Regarding experiments on more complicated architectures:**
> We tried to keep our experimental setup as simple as possible in order to clearly identify the effect of the scaling. To your point about the effect of increasing the depth, we've added 2 more preliminary experiments increasing the depth from 2 -> 4 with 1) Sin [Figure 11] and 2) ReLU activation [Figure 18]. We see the same trend of decrease in generalization performance as the scale is increased.
>
> In addition to the effect of scaling, we think a final takeaway here is the importance of our alignment measure. In practice, we feel that it is crucial to have an easily-monitorable metric that can help predict memorization, which goes beyond any particular advice such as “do not use a very large initialization scale”. We did in fact test our alignment measure on more complicated and practical models, and saw a significant correlation between its values and generalization performance, suggesting that it is a good step towards such a metric in practical settings.
>
> **Regarding generalization metrics in section 5:**
> Note that the other generalization metrics were originally formulated for example *gradients*. We did not attempt to calculate those due to a practical concern that it would be prohibitively expensive and slow to do so for larger models in Section 5. Finally, we've observed that some of the measures like Gradient Diversity and Gradient Confusion already don't correlate well with generalization performance in simpler networks like our extreme memorization study with Sin activation [Figure 2]. Regardless, we will do our best to add other generalization measures for experiments in Section 5 before the discussion period ends.

---

> > ### Comment · AnonReviewer1 · 2020-11-19
> > **After rebuttal**
> >
> > Thank you for your answer and the additional experiments.
> >
> > It would be great if you can compare with other generalization metrics. However, the new experiments looking at 4-layers networks  and the fact that you test the alignment metric on more complicated network do  address my main concern with the paper. I think the paper would be a good contribution to the venue.
> >
> > I will update my rating to reflect this.
> >
> > Thanks.

---

### Official Review · AnonReviewer4 · 2020-10-29
**Good paper with nice experiments and insights that needs cleaning up/clearer presentation**

**Rating:** 7
**Confidence:** 4

**Review:**

**Summary of paper:** A series of empirical observations are made about the influence of scale of init on generalization (in particular, that a continuum of generalization performance from random to very good can be generated by varying only the scale of init) , and these effects are explained in detail for different activation functions. The authors also propose a measure of gradient alignment which they show correlates with generalization performance

**Pros/strong points:**
 - detailed explanations for each activation function provide nice insights
 - solid experiments

**Cons/weak points:**
 - overall clarity and presentation of information is the largest weakness in my opinion, although the writing is generally good. I think it just needs a few more passes, with an eye to making sure things are accessible/understandable/flow. Could be improved substantially just with formatting/subsections or something, e.g. italicizing key insights or making sub paragraphs where each gives a particular insight
 - some small things in related work

**Summary of review + recommendation:** Overall I think this is a good paper, and could be a very good paper with some "tightening up" and clarifications. The combination of things is too much for me to recommend acceptance out of the box, but the things are relatively small and I think easy to address, and I'd be happy to increase my score.

**Detailed review and Specific questions/recommendations:**
 - unclear what "large scale training" means
 - "engendering" is an unnecessary word there
 - observations about overparameterized models should be cited, e.g. Zhang et al.  and Arpit et al.
 - Chizat & Bach further observe (not observes), same incorrect pluralization with many citations (suggest checking the whole document)
 - background work portion of the intro misses works, some poorly explained / relationship to current work not discussed, and overall feels rushed. The Related Work section does a good job mostly though. I suggest moving the 3rd p of the intro into the related work, moving the first section of it about scale of init to the first sentence of Contributions. Merging them should get you some extra space for more experiments/larger figs.
 - Related work on inits should cite lottery ticket works (e.g. Frankle et al)
 - Geiger et al reference you describe what they do but not what to take away from it
 - extreme memorization should be bolded since it's a term you're defining (and make clear if you're proposing this term and if not, where it is from), but you then  define memorization the same way you define extreme memorization, making this term ("extreme") seem unnecessary.
 - "from verylittle overfitting to perfectly memorizing the training set while making zero progress on test error" this sentence is unclear, makes it sound like "while..." applies to both of the 2 extremes. Suggest rephrasing.
 - I find 2nd bullet of contributions unclear about what is expected vs. what happens and what we learn from that
 - in 3rd bullet briefly summarize the alignment measure. I suggest using a different/more precise term for this measure (e.g. gradient alignment) since just "alignment" means so many things already
 - In "related statistics" you don't mention if Chatterjee has a measure for coherence of gradients (I skimmed that paper and it seems not, but I'm not sure). If not, then maybe calling this a measure of gradient coherence would be appropriate? I googled it quickly and it seems that coherence means something specific in linear algebra and signal processing: (https://en.wikipedia.org/wiki/Mutual_coherence_(linear_algebra), https://en.wikipedia.org/wiki/Coherence_(signal_processing)) - maybe not necessary to comment on in the paper, but if the authors are familiar with this use of the term I'd appreciate a clarification of how it relates to the mentioned measures of alignment I'd be interested
 - "related statistics" should also mention Arpit et al critical sample ratio (which does take class information into account) and comment on differences (or if it's too different to include here, I'd appreciate an explanation of why)
 - Mention computational cost of the different measures of alignment
 - homogeneity is repeatedly mentioned without explanation (just a brief 1-line would do)
 - "fix the scale" ambiguous whether this means fix in place (at a particular value) or fix as in correct
 - seems obvious to me that the scaling would affect relus (especially in the absence of bias as your experiments say); if the scale is larger, fewer values are initialized near the non-linear region of the relus, meaning there are more 'dead relus' near the beginning (which I would guess up to a point could provide regularization, but past that point would just make learning slow and even unstable). Could the authors comment on this; do you think it's correct/relevant? How does it fit with the argument about homogeneity?
 - State important equations in words as well as math for clarity and ease of reading (as is done for eq.6; make sure this is done consistently, especially important for your proposed measures of alignment).
 - Conclusion discusses the results strangely, without mentioning the actual results (e.g. "making it particularly interesting" - why/how is it interesting, what are the implications for people using sin?, "the loss function plays a crucial role" what role? what things are good for what, what should I look out for?). The conclusion should stand on its own and summarize results, not reference them in a way that requires me to have read the whole paper to understand.

---

> ### Author Response · Authors · 2020-11-17
> **Thanks for your detailed review! We have updated the paper to address all your concerns!**
>
> Thank you so much for taking the time to write such a detailed and helpful review of our paper. Applying your detailed suggestions has indeed improved the quality of our paper. We've updated our paper to incorporate all your comments. In light of that, we really appreciate it if you would consider increasing your score to “accept”.
>
> Following are the responses to the specific questions/concerns raised:
>
> **Main concern (regarding presentation):**
> We have updated our paper to improve the presentation. We have further addressed all the specific concerns raised including reorganising the intro/related work, fixing pluralization, making conclusion self-contained and more specific, adding relevant citations, editing ambiguous sentences to be more precise, adding more context along with the equations in order to motivate them, making changes to contributions so that they are clearer, adding computational cost of measures etc.
>
>
> **Regarding some of the related work:**
>
> Regarding coherence: yes! we do refer to the work of Chatterjee in our related section. Like you said, we could not find a measure which we could compare in our work. Although, we feel that coherence as explained in their work is slightly different than what we wanted to measure. More specifically, in their work 1) the gradients are not normalized and 2) it doesn't take true class information into account. We feel that cosine stiffness is more similar to our gradient alignment measure but suffers in computational complexity and in scenarios where example gradients have widely varying norms.
>
> Regarding Critical Sample Ratio from Arpit et al: you are right that it does take class information into consideration but it seems to be dependent on the predicted label (i.e. argmax of the class scores) and not the true label. In contrast, we take the true label of the training data point into account. More generally, we feel that CSR was aimed at measuring the density of decision boundaries which seems slightly orthogonal to our goal of measuring the degree of agreement between example gradients. It is not immediately clear to us how we can reformulate CSR as a function of example gradients to make an equivalence. But please let us know if we are missing anything.
>
> **Regarding scaling affecting ReLU**
> This is a good point - you are right that in any given example, dead ReLUs might be pushed further into the “dead region” by increasing the scale. However, we note that the positive-homogeneity implies that the *number* of dead ReLUs at the beginning of training is actually invariant since the output will simply be scaled by a positive constant. Moreover, after ReLU layers, increasing the scale only changes the scale of the logit values and not the relative ordering. Further, our feeling is that the issue of making dead ReLUs further from zero is ameliorated by the gradient computed over a batch of examples, for which ReLUs will not always be off (if a unit is “dead” for ALL examples, then it would have been dead and not pass on gradients regardless of the scaling). Thus, one might imagine that if this were the problem, it would be fixable by changing scaling the learning rate appropriately to traverse the increased distance to the ReLU boundary in the same amount of time. However, changing the learning rate alone is unable to prevent memorization behavior in our experience (see also our response to Reviewer 3). Further, note that the problem of dead ReLUs would be present in the training set as well, but the net is perfectly capable of memorizing the training set, suggesting that this issue is not the primary problem. Finally, similar to ReLU, we also see drops in accuracy while increasing the scale of initialization in the case of linear activation where this problem is not present altogether.
>
> We have fixed all the other issues you pointed out in the paper. Please take a look and let us know if we missed anything or any other relevant papers we should cite. Thank you!

---

> > ### Comment · AnonReviewer4 · 2020-11-22
> > **Last remaining issue: precision of language around "extreme memorization" vs more general "memorization"**
> >
> > Thanks for taking the time to thoroughly address concerns! I think the organization and especially the contributions section and related work are much improved.
> >
> > A couple remaining comments/suggestions after reading the revision:
> > * "with sin being the most extreme": extreme in what? suggest changing to "demonstrating the most extreme memorization" or something similar that describes a particular result instead of a vague qualitative description.
> > * I suggest changing the sentence on the first page to: "refer to this behavior as extreme memorization, distinguished from the more general category of memorization (which could describe near-random-chance behaviour at relatively-low accuracy).", and then be consistent about using extreme memorization only when you really mean it specifically.   Importantly about this point, the next mention of "perfectly memorizing the training set" seems fine, because your experiments indeed vary the degree of memorization, with "extreme memorization" (i.e. random-chance test-set performance combined with perfect memorization) happening only at "one end" of the experiments. But I think it's important to describe this clearly and use your terms correctly and precisely, especially when you are introducing them. Don't use terms like "complete memorization" that are ambiguous thereafter. Since you have some space, I also suggest adding a line with this definition to the section titled "Extreme Memorization" (that is where I would look if I wanted to know the definition of this term).
> >
> > Assuming these small but important ambiguities in language will be addressed, I have increased my score and recommend acceptance, I think this is an interesting paper and a good contribution.

---

> > > ### Author Response · Authors · 2020-11-24
> > > **Thank you! New revision addresses both of your minor concerns!**
> > >
> > > Thank you very much for increasing the score and pointing out the remaining issues!
> > >
> > > We have addressed both of your minor concerns in our new revision, namely
> > >
> > > - Changing "with sin being the most extreme" to "demonstrating extreme memorization" and
> > > - Adding your suggested edits, removing reference to vague terms like "complete memorization" and reiterating the definition of extreme memorization at the start of Section 3.
> > >
> > > We will also make sure to keep the crux of your advice in mind while making further passes. Please let us know if you have any other suggestions or if we have missed anything.

---

### Author Response · Authors · 2020-11-17
**New version incorporating feedback from all reviews**

We thanks all reviewers for their feedback! We have uploaded a new version with suggested changes and new experiment results as mentioned in our official responses to the reviews.

---

### Comment · ~Olivier_Grisel2 · 2021-01-28
**Relation to Shattered Gradients by Balduzzi et al.**

Interesting work. I believe this is quite related to the empirical analysis in:

The Shattered Gradients Problem: If resnets are the answer, then what is the question?
David Balduzzi, Marcus Frean, Lennox Leary, JP Lewis, Kurt Wan-Duo Ma, Brian McWilliams,
ICML 2017,
https://arxiv.org/abs/1702.08591

Although the Shattered Gradients gradient paper focuses on deep networks (with ~30+ layers) and analyzes the covariance structure of the gradients w.r.t. the inputs and they do not study the scale of the init but rather whether the networks is close to linear at init time (with the Looks Linear init). Still I suspect that the "effective rank" measure they use to quantify the problem of shattered gradients (of the loss w.r.t. network inputs) is possibly very related to the (parameter) gradient alignment measure introduced in this work (Extreme Memorization).

---

> ### Author Response · Authors · 2021-01-29
> **Thank you for the pointer and suggestions!**
>
> Thank you for taking a look at our work and also pointing us to the Shattered Gradients by Balduzzi et al reference!
>
> We fully agree with you that the 'effective rank' measure is quite related to the alignment measure and we will investigate it more in our context. We will also add Balduzzi et al as a reference in our related work section of our camera ready revision. Besides that, as you correctly pointed out, we feel that both papers are quite different. In addition to the things you pointed out, our main focus is how *generalization* gets affected by misalignment in the gradients and they seem to focus on *optimization* ability, both topics being super important to study in our opinion.

---

> > ### Comment · ~Olivier_Grisel2 · 2021-02-01
> >
> > Indeed, your point on the focus on optimization ability of the shattered gradients paper is very valid.

---

### Decision · Program_Chairs · 2021-01-07
**Final Decision**

**Decision:**

Accept (Poster)

**Comment:**

This paper provides a clear and useful empirical study of how the initialization scale and activations function affects the generalization capability of neural networks. Previous works showing the effect of the initialization scale (Chizat and Bach (2018), Geiger et al. (2019), Woodorth et al. (2020)) had a more limited set of experiments. Moreover, here an extreme case is shown, wherewith sin activation function no generalization is possible at a large init scale (there the kernel regime is useless for generalization since the hidden layer output becomes very sensitive to any small perturbation in the input). Lastly, two alignment measures are suggested, which are correlated with the generalization across several architectures and initialization scales.

All the reviewers argued for acceptance, and one strongly so. I agree that the paper is sufficiently interesting and clear to be accepted. However, despite the high scores, I only recommend a poster and not spotlight/oral: I think the novelty of the empirical study is not groundbreaking, given the experiments in previous works, and the usefulness of the suggested measures are not completely clear without a thorough comparison against previously suggested measures.